



# The importance of burning conditions on the composition of domestic biomass burning organic aerosol and the impact of atmospheric aging

Rhianna L. Evans[1], Daniel J. Bryant[1], Aristeidis Voliotis[2,3], Dawei Hu[2], HuiHui Wu[2], Sara Aisyah Syafira[2], Osayomwanbor E. Oghama[2], Gordon McFiggans[2], Jacqueline F. Hamilton[1,4], and Andrew R. Rickard[1,4]

[1]Wolfson Atmospheric Chemistry Laboratories, Department of Chemistry, University of York, York, YO10 5DD
[2]Centre for Atmospheric Science, Department of Earth and Environmental Sciences, School of Natural Sciences, University of Manchester, Manchester, M13 9PL
[3]National Centre for Atmospheric Science, University of Manchester, Manchester, M13 9PL
[4]National Centre for Atmospheric Science, University of York, York, YO10 5DD

**Correspondence:** Jacqueline F. Hamilton (jacqui.hamilton@york.ac.uk) and Andrew R. Rickard (andrew.rickard@york.ac.uk)

**Abstract.** Domestic biomass burning is a significant source of organic aerosol (OA) to the atmosphere however the understanding of OA composition under different burning conditions and after oxidation is largely unknown. Compositional analysis of OA is often limited by the lack of analytical standards available for quantification, however, semi-quantitative non-target analysis (NTA) can overcome these limitations by enabling the detection of thousands of compounds and quantification via surrogate

standards. A series of controlled burn experiments were conducted at the Manchester Aerosol Chamber to investigate domestic biomass burning OA (BBOA) under different burning conditions and the impact of atmospheric aging. Insights into the chemical composition of fresh and aged OA from flaming dominated and smouldering dominated combustion were obtained via a newly developed semi-quantitative NTA approach using ultra-high-performance liquid chromatography high-resolution mass spectrometry. Aerosol from smouldering dominated burns contained significant organic carbon content whereas under

flaming dominated conditions was primarily black carbon. The detectable OA mass from both conditions was dominated by oxygenated compounds (CHO) ($\approx$ 90 %) with smaller contributions from organonitrogen species. Primary OA (POA) had a high concentration of $C_8$-$C_{17}$ CHO compounds with both burns exhibiting a peak between $C_8$-$C_{11}$. However, flaming dominated POA exhibited a greater contribution of $C_{13}$-$C_{17}$ CHO species. More than 50 % of the CHO mass in POA was determined as aromatic by the aromaticity index, largely in the form of functionalised monoaromatic compounds. After aging the aromatic

contribution to the total CHO mass decreased with a greater loss for smouldering (-53%) than flaming (-16 %) due to the increased reduction of polyaromatic compounds under smouldering conditions. The O:C ratios of the aged OA from flaming and smouldering were consistent with those from the oxidation of aromatic compounds (0.57 - 1.00), suggesting that compositional changes upon aging were driven by the oxidation of aromatic compounds and the loss of aromaticity. However, there was a greater probability of O:C ratios $\geq$ 0.8 in aged smouldering OA indicating the presence of more oxidised species. This

study presents the first detailed compositional analysis of domestic BBOA using a semi-quantitative NTA methodology and





demonstrates compositional changes between burn phase and after aging may have important consequences for exposure to such emissions in residential settings.

## 1 Introduction

Biomass burning (BB) encompasses a range of combustion processes such as wildfires, agricultural burning, and domestic combustion of solid fuels or referred to herein, domestic BB. Biomass burning is one of the largest sources of organic aerosol (OA) and trace gases to the atmosphere, emitting approximately 62 Tg yr$^{-1}$, 77 Tg yr$^{-1}$ and 19 Tg yr$^{-1}$ of volatile organic compounds (VOCs), particulate matter less than 2.5 $\mu$m in diameter (PM$_{2.5}$) and nitrogen oxides (NO$_x$) respectively to the atmosphere (Andreae, 2019). Biomass burning VOCs (BBVOCs) can oxidise in the atmosphere leading to the production of
secondary organic aerosol (SOA) which is a major component of PM$_{2.5}$. With respect to human health, PM$_{2.5}$ is a highly toxic air pollutant as fine particles can be inhaled deep into the respiratory tract (Kampa and Castanas, 2008). Approximately 2 billion people globally use solid fuels for heating and cooking (World Bank, 2024) representing a chronic exposure to poor air quality and annually solid fuel combustion is responsible for 3.2 million deaths worldwide (World Health Organisation (WHO), 2022). However it was estimated that 20 % of the global annual deaths caused by PM$_{2.5}$ could be avoided by eliminating domestic
BB (McDuffie et al., 2021). In the UK, approximately 8 % of the population burn wood indoors (Department for Environment Food & Rural Affairs (DEFRA), 2020) and emissions from solid fuels comprised approximately 26% of the total primary OA (POA) in London during cold weather conditions consistent with domestic heating activity (Allan et al., 2010). Domestic BB is predicted to grow due to numerous energy crises in recent years and as a cheaper alternative to gas (International Energy Agency, 2022) in the rising costs of living. However, emissions from wood burning are highly dependent on combustion
conditions, the fuel burnt, and the stove appliance used (Andrew Price-Allison, 2022).

A full burn cycle consists of multiple stages i) ignition, ii) flaming combustion and iii) smouldering combustion (Andreae and Merlet, 2001). During flaming the lignocellulosic biomass is partially or completely burned and char is produced, typically occurring at high temperatures. Whereas smouldering occurs during the latter stages of the burn cycle at lower temperatures, starting once all the combustible volatile fuel is consumed and the oxidation of char begins (Andreae and Merlet, 2001). Due
to these unique conditions a characteristic mixture of VOC emissions arises at each stage from temperature dependent pyrolysis mechanisms at varying abundances (Andreae and Merlet, 2001; Czech et al., 2016; Liu et al., 2017; Stewart et al., 2021). For instance, Czech et al. (2016) observed the greatest emissions from the ignition phase followed by ember (smouldering) and stable burn (flaming) phases. Previously the change in emissions between burn phase was identified using positive matrix factorisation which separated BBVOC emissions into two factors; low and high temperature combustion. Low temper-
ature combustion contained more oxygenated aromatics and furanic compounds in agreement with particle phases enriched in oxygenated organic compounds from smouldering combustion (Sekimoto et al., 2018; Weimer et al., 2008). In contrast, BB-



VOCs from high temperature combustion consisted of polyaromatic hydrocarbons (PAHs), terpenes and aliphatic unsaturated hydrocarbons (Sekimoto et al., 2018; Stefenelli et al., 2019).

Ultra-high-performance liquid chromatography coupled to electrospray ionisation high-resolution mass spectrometry (UHPLC-ESI-HRMS) is a valuable technique for studying the composition of OA enabling the detection of thousands of compounds and separation of isomeric species. Various tracer species from biomass burning OA (BBOA) have been previously identified using UHPLC-HRMS (e.g., Claeys et al., 2012; Kourtchev et al., 2016; Budisulistiorini et al., 2017; Iinuma et al., 2007; Capes et al., 2008; Daellenbach et al., 2019; Brege et al., 2018; Wang et al., 2017b; Smith et al., 2020; Zangrando et al., 2013), but most commonly consisted of levoglucosan and nitroaromatic compounds (NACs) (Claeys et al., 2012; Kourtchev et al., 2016; Budisulistiorini et al., 2017; Iinuma et al., 2010; Piot et al., 2012; Kitanovski et al., 2012; Li et al., 2017). Biomass burning plumes are ideal conditions for NAC formation due to high emissions of $NO_x$ and aromatic VOCs from lignin degradation which is primarily comprised of 3 aromatic alcohol units; coumaryl, sinapyl and coniferyl alcohol (Simoneit et al., 1993). In particular NACs such as nitrophenols are widely adopted as tracers due to strong correlations with levoglucosan (Iinuma et al., 2010; Jiang et al., 2020; Cai et al., 2022) and depending on the fuel type emission factors range between 1.4 - 31 mg kg$^{-1}$ (Iinuma et al., 2007, 2010; Claeys et al., 2012; Kourtchev et al., 2016). These compounds also have important climatic impacts by contributing to brown carbon (BrC) hence their extensive study in the wider literature (Fleming et al., 2020; Zhou et al., 2022; Wang et al., 2020; Lin et al., 2016; Gilardoni et al., 2016). However, by selecting a small number of compounds to analyse, limited compositional information can be obtained by targeted approaches, for instance, Pereira et al. (2021) estimated only 1.1% of the mass of an ambient OA filter could be quantified via a targeted approach using 60 authentic standards.

Non-target analysis (NTA) can overcome these limitations by enabling chemical information, such as molecular formula, of all detected analytes within a complex mass spectral output to be rapidly obtained. For example, NTA enabled the identification of 190 NACs in PM$_{2.5}$ from Beijing, with a third attributed to biomass burning (Wang et al., 2021). Furthermore, species associated with lignin pyrolysis such as vanillin, coniferaldehyde and benzoic acid and sugars including levoglucosan, sucrose and fructose from cellulose degradation were previously identified in BBOA via NTA approaches (Simoneit, 2002; Smith et al., 2020, 2009). However, due to the lack of commercially available authentic standards many previous NTA studies used limited metrics such as number of molecular formulas or peak area to estimate relative abundance (eg., Dzepina et al., 2015; Wang et al., 2020; Brege et al., 2021; Smith et al., 2009; Pereira et al., 2021; Brege et al., 2018; Herrera-Lopez et al., 2014). The lack of standardised metrics to estimate abundance can lead to differences in inferred composition. Using peak area, Wang et al. (2020) and Brege et al. (2018) observed a large quantity of organonitrogen compounds (CHON) in OA during periods influenced by biomass burning, which were attributed to NACs. In contrast, using the number of formulas showed a greater contribution of oxygenated organic compounds (CHO) in BBOA (Brege et al., 2021; Dzepina et al., 2015; Smith et al., 2009). For example, at the Pico Mountain Observatory in the North Atlantic, analysis of PM$_{2.5}$ samples showed CHO compounds accounted for 70 % of the molecular assignments in air masses influenced by wildfires (Dzepina et al., 2015) and Smith et al. (2009) observed CHO compounds represented 80-90% of all detected mass spectral features in BBOA.

Whilst the number of formulas can provide some information on sample complexity it is not quantitative for concentration and although peak area is often considered as quantitative neither of these approaches accounts for differences in ionisation



efficiency (IE) between different species. IE is a measure of the ability of a species to ionise within an ESI source and is highly structural specific, varying by multiple orders of magnitude between different compounds including structural isomers (Oss et al., 2010; Liigand et al., 2021). For instance, across isomers of $C_7H_7NO_3$ (methyl-nitrophenol) IE can vary by 3 orders of magnitude (Evans et al., 2024) (*in review*). Therefore improved NTA methodology incorporates quantification into the workflow. These approaches have largely quantified compounds with known structures through the use of response factor (RF) models which predict the ionisation efficiency of a known compound relative to a reference compound (Bryant et al., 2023; Liigand et al., 2021; Mayhew et al., 2020). However, the quantification of unidentified compounds remains a challenge. Sepman et al. (2023) recently developed a model which used fragmentation spectra ($MS^2$) to obtain molecular fingerprints of unknown species for the prediction of RF. Often NTA UHPLC-HRMS approaches use data-dependent $MS^2$ ($ddMS^2$) to obtain higher quality $MS^2$ (Guo and Huan, 2020) which means only a certain number of species are selected for fragmentation in each scan. Therefore relying on $MS^2$ for quantification means some compositional information is lost. For example, using a NTA workflow Wang et al. (2022) observed only 39% of detected compounds had $MS^2$ spectra using $ddMS^2$. Alternative methods for quantification of unknowns without $MS^2$ involves semi-quantification using a singular structurally similar surrogate standard which is assumed to have a similar ionisation efficiency to the target compound (Kim et al., 2023a; Kruve et al., 2021; Pieke et al., 2017; Rattanavaraha et al., 2016; Wang et al., 2023). A recent methodology outlined in Evans et al. (2024) (*in review*) enabled the quantification of all unidentified species by using average ionisation efficiencies of surrogate standards which elute within the same retention time window as the unknown analytes. The coupling of retention time and chemical group enabled quantification by chemically and structurally similar compounds and therefore more reliable estimates of concentration with uncertainties. This semi-quantitative methodology had a relatively low quantification error of 1.52 times compared to quantification using authentic standards across 27 identified compounds in the domestic BBOA dataset characterised in this study. Thereby enabling more reliable estimates of the relative quantities of all detected species in NTA workflows without knowing the structural identity (Evans et al., 2024) (*in review*).

In this study, chamber experiments were conducted at the Manchester Aerosol Chamber (MAC) to investigate the effect of combustion conditions on the chemical composition of fresh OA emitted from domestic wood burning and the subsequent aged OA. Using a domestic wood burning stove, emissions during burn phases dominated by smouldering or flaming were sampled and subsequently photochemically aged inside an atmospheric simulation chamber. UHPLC-HRMS in negative mode ESI was used to analyse the chemical composition of OA formed from different burn phases in fresh and aged emissions. Overall this study represents the first molecular level semi-quantitative NTA of domestic BBOA to improve our understanding of the prominent species contributing to domestic BBOA under different burn phases and the compositional changes occurring after atmospheric aging. These findings aim to aid future policy on the mitigation of poor air quality, climatic impacts and the harmful health effects from domestic BB.



## 2 Methodology

### 2.1 Controlled burn chamber experiments

#### 2.1.1 Design

Controlled burn chamber experiments were conducted over two campaigns during April 2022 and September 2022 at the Manchester Aerosol Chamber (MAC) located at the University of Manchester, UK. The experiments aimed to investigate the aerosol composition under different burning conditions. $PM_{2.5}$ filter samples were taken for detailed offline composition analysis and a large suite of online instrumentation measured aerosol composition, aerosol physical properties and trace gases,

including carbon monoxide (CO), carbon dioxide ($CO_2$), ozone ($O_3$) and $NO_x$. The MAC has previously been described in detail elsewhere (Shao et al., 2022). However, in brief the MAC consists of an enclosed and suspended 18 $m^3$ fluorinated ethylene Teflon bag supported by 3 rectangular aluminium frames, where the outer frames move freely allowing the bag to expand or contract when filling or emptying the chamber. The chamber was illuminated using two 6kw xenon arc lamps with quartz fibre glass filters and 4 rows of halogen lamps (64 bulbs) to simulate atmospheric solar wavelengths. Purified dry air

was supplied to the chamber by passing laboratory air through a 3-phase blower and 3 filters comprising i) purafil/charcoal mixture, ii) charcoal and iii) HEPA. The chamber had automated fill/flush cycles before and after experiments, detailed in Shao et al. (2022), to reduce the chamber background signal and was cleaned overnight with high concentrations of $O_3$ ($\approx$ 1ppm) to oxidise any residual species. A harsher cleaning programme was performed once a week by illuminating the chamber for 4-5 hours under high $O_3$ concentrations ($\approx$ 1ppm). During the controlled burn campaigns multiple background experiments were

conducted whereby a clean chamber, ie. no added smoke from the stove, was irradiated with light for 6 hours. After this time, a filter sample was collected of the chamber air via the flush line at approximately 3 $m^3$ $min^{-1}$ for 5 minutes.

For the controlled burn experiments, hardwood (Beech), which typically provides more heat and burn for longer than softwood, was burnt in a Ecodesign stove (Esse Model 175 F) to represent a typical domestic fuel in the UK. The emissions from the burn were sampled during flaming or smouldering phases. However given the nature of a burn which will be composed of

both flaming and smouldering processes it is difficult to separate distinct phases therefore the burns are referred to herein as "flaming dominated" and "smouldering dominated". The catalytic filter within the stove, which would enable the "particulate reburn" technology to reduce particulate emissions beyond that of the UK Ecodesign requirements, was removed to replicate more conventional stoves in the UK market. A filter of the POA was taken from the flue of the wood burner at 2 L $min^{-1}$ for 5 minutes for offline chemical composition analysis before injection of the smoke into the chamber. The wood smoke from the

flue derived from either smouldering or flaming dominated phases (2 L $min^{-1}$) was then diluted with a flow of compressed air (2 L $min^{-1}$) before injection into the chamber using an eDiluter (eDiluter Pro, Dekati, Finland). The smouldering dominated phase was controlled by allowing the wood to be consumed by flames and subsequently closing the stove ventilation to reduce the presence of oxygen in the stove. Injection into the chamber started when there was a lack of visible flames. No additional reactants were injected into the chamber. The injection of the smoke proceeded until the total particle mass concentration mea-

sured in the chamber was twice the target concentration at half the chamber volume. Then the final addition of scrubbed particle





**Table 1.** List of the OA samples used in this study and the initial conditions at the start of the aging period

| Experiment date | Conditions | Sample ID | Aging period / hrs | PM concentration / $\mu g\ m^{-3}$ | NO:NO$_2$ | OC:BC | CO:CO$_2$ |
|---|---|---|---|---|---|---|---|
| 21/04/2022 | Flaming light aged | FL_AGED_1 | 5:50 | 243.6 | 1.94 | 0.32 | 0.039 |
| 26/04/2022 | Smouldering light aged | SM_AGED | 6:05 | 213.6 | 1.81 | 406.3 | 0.287 |
| 28/04/2022 | Flaming light aged | FL_AGED_2 | 6:05 | 153.4 | 3.74 | 0.21 | 0.075 |
| 30/08/2022 | Flaming fresh flue | FL_FRESH | - | - | - | - | - |
| 31/08/2022 | Smouldering fresh flue | SM_FRESH | - | - | - | - | - |

free air into the bag achieved the target concentration of around 200 $\mu gm^{-3}$ at full chamber volume. Following injection, 40-60 minutes of background data was collected to allow instrumentation with long cycle times to obtain several cycles of data. After this period the chamber was irradiated to produce OH radicals for photo-oxidation. The smoke was aged for approximately 6 hours before sampling the chamber via the flush line for 4 minutes to collect the aged aerosol onto a pre-baked Quartz filter for
offline chemical composition analysis. The exact flow rate of the vacuum line for collection was not directly measured however the MAC can be entirely flushed from full in approximately 6 minutes therefore the flow rate is approximated as 3 m$^3$ min$^{-1}$ (Shao et al., 2022). Quartz filters (Whatman QMA, 47mm) used for sample collection were individually wrapped in foil and pre-baked at 500 °C for 5 hours prior to use. After collection the filters were wrapped in the pre-baked foil, transported on ice to the University of York and finally stored at -20 °C before offline analysis.
Each aging experiment was repeated once resulting in two sample types per burn phase, fresh and light aged, with one filter for the fresh emissions and two filters for the aged experiments. However, as previously stated in practice the fire can be undergoing flaming and smouldering phases at the same time meaning it can be difficult to exactly separate a singular phase into the chamber. As such one of the repeats aiming to capture the smouldering dominated phase was characterised as an intermediary burn with increased flaming characteristics, ie. high CO$_2$ concentrations, but also exhibiting high ratios
of organic:black carbon (OC:BC) associated with smouldering. This experiment is not discussed further in this study. The experiments, their sample ID referred to in this study and initial concentrations of gas and particle phase species before aging are presented in Table 1. Note that fresh aerosol samples were taken from the September campaign as the April campaign only sampled aerosol from the chamber.

### 2.1.2 Online Instrumentation

The experiments used a variety of online instrumentation to monitor the evolving aerosol and gaseous composition throughout the photo-oxidation of the wood burning smoke inside the MAC. Non-refractory PM$_1$ composition was measured via a high-resolution-time-of-flight aerosol mass spectrometer (HR-ToF-AMS) enabling real-time measurements of NH$_4^+$, NO$_3^-$, Cl$^-$, SO$_4^-$ and the organic fraction. The extent of oxidation could be monitored using the fraction of the 44 *m/z* fragment compared





to the total organic fraction ($f_{44}$) with higher $f_{44}$ levels associated with more oxygenated organic aerosol. Whilst the signal at

*m/z* 60 has been associated as a fragment from biomass burning tracers such as levoglucosan and other structurally similar

sugars. Therefore, the degradation of the wood smoke throughout photo-oxidation was also monitored using the fraction of *m/z*

to the total organic fraction ($f_{60}$). Additionally, particle concentrations were measured using a Scanning Mobility Particle

Sizer (SMPS) across a size range of 10-700 nm and measurements of black carbon mass and coating thickness were obtained

from a Single Particle Soot Photometer (SP2).

**2.1.3  Offline Instrumentation**

Prior to offline analysis the filters were extracted based on the method used in Bryant et al. (2023). The 47 mm quartz filters

were cut into 1 cm$^2$ pieces, placed in a 20 mL glass vial and 10 mL of methanol (LC-MS Optima Grade) was added. For

fresh aerosol samples half a 47mm filter was used due to the higher aerosol mass loading. The resulting 10 mL solution was

sonicated for 45 minutes, using ice packs to lower the temperature of the water bath. The methanol extract was transferred to a

second 20 mL glass vial using a 0.22 $\mu$m syringe filter (Milipore) then dried using a Genevac vacuum solvent evaporator. The

sample was reconstituted in 200 $\mu$L 90:10 H$_2$O (LC-MS Optima Grade): MeOH (LC-MS Optima Grade) for UHPLC-HRMS

analysis.

The offline filters were characterised at the University of York using an Ultimate 3000 UHPLC (Thermo Scientific, USA)

coupled to a Q Exactive Orbitrap MS (Thermo Fisher Scientific, USA) with heated electrospray ionisation (HESI) enabling

high resolution and detailed chemical information to be obtained. Compound separation was achieved using a reversed phase

C$_{18}$ 2.6 $\mu$m x 2.1 mm x 10 mm Accucore column held at 40 °C. The mobile phase consisted of 0.1 % (v/v %) formic acid

(Acros Organics) in water (A, LC-MS Optima Grade) and methanol (B, LC-MS Optima Grade). A gradient elution was used,

starting at 90 % (A) with a 1 minute post injection hold, decreasing to 10 % (A) at 26 minutes before returning to the starting

conditions at 28 minutes. A final 2 minute hold at 10 % (A) allowed to the column to re-equilibrate. The flow rate was set

to 0.3 mL min$^{-1}$and prior to analysis samples were stored in an autosampler tray at 4 °C. The injection volume was set to 4

$\mu$L, however injection volumes up to 10 $\mu$L were used for lower concentration samples. The HESI was operated under the

following conditions: a spray voltage of 4 kV, a capillary and auxiliary gas temperature of 320 °C, a sheath gas flow rate of 45

(arb.) and an auxiliary gas flow rate of 10 (arb.) Spectra were acquired in negative and positive mode using ddMS$^2$ however this

study only considers those acquired in negative mode. This is because of the greater sensitivity of positive mode meaning there

are a greater number of compound functionalities can be detected which requires a significant amount of analytical standards to

estimate IE. Whereas the negative mode is more selective requiring analytical standards of fewer functionalities to develop the

semi-quantitative methodology. The scan range was set to a mass-to-charge ratio (*m/z*) of 85 to 750, with a mass resolution of

140,000. Tandem mass spectrometry was performed using a higher collision dissociation with a stepped normalised collision

energy of 10, 20 and 45. In each scan the 10 most abundant species were selected for MS$^2$ fragmentation. The wood burning

samples were analysed once with solvent blanks and chamber blanks analysed at the start of the sequence for blank subtraction

in post-processing.





## 2.2 Semi-quantitative non-target analysis

Spectra were acquired from XCalibur 4.3 (Thermo Scientific, USA) and analysed using a semi-quantitative non-target workflow developed by Evans et al. (2024) (*in review*) for analytes detected in negative mode. In brief this method uses a non-target
workflow developed in MZmine 2.53 and MZmine 3.9.0 software to detect features, assign molecular formulas, and identify compounds via a spectral library search. The post processing proceeds in the following steps i) selection of the best predicted formula, ii) blank subtraction and iii) removal of data which ionised better (ie.larger peak area) in positive mode ESI and iv) semi-quantification of all detected analytes. For i) the formula with the lowest mass tolerance in ppm was chosen as the "best" formula if within the elemental ratios: 0.5 < H:C < 3, 0.05 < O:C < 2, N:C < 1, S:C < 0.5 and Cl:C < 0.2. For ii) common species
detected in the sample and filter blank or chamber blank were removed if the sample-to-blank signal was < 10 and species in the wood burning samples were removed if the signal-to-noise ratio was < 3. In the final step iv) quantification is achieved via closely eluting surrogate standards for each chemical group (ie CHO, CHON, etc.). The acquired chromatogram from the UHPLC-HRMS method was split into retention time windows and authentic standards and sample analytes were assigned to a window based on their retention time and chemical group. For CHO the number of standards allowed retention time windows
of 1 minute from 0 - 14 minutes and windows of 2 minutes from 16 - 20 minutes resulting in 17 retention time windows. For CHON retention time windows range between 2-3 minutes due to the lower number of available standards resulting in 8 retention time windows. Overall this methodology uses 110 standards across the retention time windows to derive average scaling factors used in quantification. A scaling factor was obtained for each retention time window by calculating the median calibration slope across the authentic standards present within each window. For CHO species these slopes were predominantly
derived from organic acids and for CHON species the method uses nitroaromatic standards as these are compounds likely to be observed in BBOA and are selective to negative mode ESI. Whilst for sulfur containing species (eg. CHOS, CHOSN etc) due to the lack of authentic standards a single compound, camphorsulfonic acid, is used for quantification. For the non-identified compounds, quantification is achieved via the scaling factor for the corresponding retention time window. Whereas for the compounds identified by the spectral library, quantification is achieved using an authentic standard.

## 3 Results

### 3.1 Insights into the oxidation of organic aerosol from online measurements

Emissions from domestic BB under different burning conditions, ie. flaming dominated or smouldering dominated, were photo-oxidised inside the MAC to observe the impact of atmospheric aging on the chemical composition of domestic BBOA. The particulate emissions from flaming dominated or smouldering dominated burn phases show that flaming is primarily formed of
BC whereas smouldering shows significantly higher concentrations of OC (Fig. A1) which could impact the particle morphology (Leskinen et al., 2014) and thereby the aging of OA. The photo-oxidation of domestic BBOA was monitored in real time with an AMS to gain insight into the evolving organic fraction of non refractory $PM_1$ alongside instrumentation to measure the concentrations of trace gases such as nitrogen oxides ($NO_x = NO + NO_2$) (see Fig. A2a). Figure 1 shows the relationship





between $f_{44}$, representing oxidised components, and $f_{60}$, indicative of the levoglucosan-like species typically used as tracers of
BB, over the course of the experiment and generally exhibits a negative trend of $f_{60}$ with increasing $f_{44}$. This trend therefore in-
dicates the components of the fresh BB emissions are undergoing various aging processes, due to the reduction in $f_{60}$, including
chemical oxidation to form more oxidised species as indicated by the increase in $f_{44}$. However the negative correlation with $f_{60}$
shown in Fig. 1 varies between the emissions from flaming and smouldering dominated experiments indicating the composition
and atmospheric aging of OA is impacted by the burning conditions. This was similarly observed in a study investigating solid
fuel emissions in London which associated two factors derived by positive matrix factorisation with two distinct $f_{44}$ and $f_{60}$
spaces arising from differences in burning phase or fuel type (Young et al., 2015). The increase in $f_{44}$ ranges between 0.065 -
0.08 depending on the burn phase which is similar to the increase in $f_{44}$ observed by Brege et al. (2018) (+0.055) between fresh
and aged ambient BBOA. For the flaming dominated emissions the reduction in $f_{60}$ is considerably less (-0.0006-0.0013) than
the smouldering dominated phase (-0.025). This could be a result of reduced levoglucosan emissions during the flaming phase
(Lee et al., 2010; Shafizadeh, 1982). The observed range of $f_{44}$ and $f_{60}$ values in Fig. 1 are in agreement with previous BBOA
studies (Cubison et al., 2011; Jolleys et al., 2015; Adler et al., 2011) which are typically situated within the triangular bounds
of $f_{44}$ (0.05-0.25) vs $f_{60}$ (0.01-0.04) observed by Cubison et al. (2011). Overall these results indicate the oxidation of POA
from fresh domestic BB emissions to form oxidised POA (oPOA) and SOA. Multiple aging processes such as the evaporation
of semivolatile species, condensation of oxidised vapours and the photochemical formation of SOA could contribute to the
increased $f_{44}$ in the aged OA.

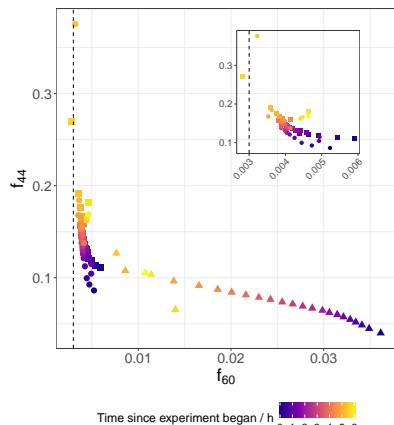

**Figure 1.** Online AMS measurements of $f_{44}$ and $f_{60}$ for flaming dominated (circle and square points) and smouldering dominated (triangle
points) burn phase during the aging experiments, averaged to 15 minute intervals. Sample IDs are given by the point shape. The inset plot
displays the zoomed in flaming data. The colour of the points represents the time since the experiment began. The vertical dashed line at $f_{60}$
$\approx 0.3\%$ represents non biomass burning influenced OA ((Cubison et al., 2011))

The concentration of $NO_x$ was greater from the flaming dominated emissions compared to the smouldering dominated phase
as shown in Fig. A2a which is as expected and in agreement with previous flaming phase observations (Andreae and Merlet,



2001; Andreae, 2019; Gilman et al., 2015; Roberts et al., 2020). The ratio of NO:NO$_2$ was used to infer the photochemical

conversion of NO to NO$_2$ (Fig. A2b) indicative of the oxidation of VOCs, which results in the production of secondary species

such as O$_3$ and SOA. NO:NO$_2$ ratios were initially higher under the flaming dominated conditions compared to smouldering

dominated emissions which is consistent with NO as the end product of nitrogen oxidation at higher temperatures (Lobert and

Warnatz, 1993) and flaming as a more efficient burn phase. In the smouldering phase, NO$_2$ emission can account for up to 40%

of the total NO$_x$ (Lobert and Warnatz, 1993) resulting in a lower NO:NO$_2$ ratio which is in agreement with the observed NO$_2$

contribution (36%) to the total initial NO$_x$ during the smouldering dominated phase in this study. After aging the NO:NO$_2$

ratios in both burn phases converge to a steady state ($\approx 0.2$) in Fig. A2b as NO is photochemically converted to NO$_2$, through

the reaction with peroxy radicals formed from BBVOC oxidation. As observed in Fig. 1 this indicates aged OA will contain a

mixture of POA, oxidised POA (oPOA) and SOA. Ambient BB plumes report NO:NO$_2$ ratios up to 3-5 (Jenkins et al., 1991;

Oppenheimer et al., 2004) which are greater than those presented in Table 1 suggesting an increased proportion of NO$_2$ in this

work. This was observed in a previous chamber study and attributed to the presence of O$_3$ within a dark chamber (Delmas et al.,

1995). However, due to interference from conjugated VOCs the concentrations of O$_3$ inside the chamber cannot be accurately

quantified from the UV measurements. Overall online measurements show that the burn phase influences the initial conditions

inside the chamber including trace gas concentration and OA composition which can lead to differences in the atmospheric

aging of OA.

### 3.2 Non-target analysis of organic aerosol derived from different burn phases

The NTA methodology described in Sect. 2.2 enables large quantities of chemical information to be obtained for all detected

compounds, including those with unknown structural identities (Evans et al., 2024) (*in review*). Table A1 shows some com-

monly used aerosol metrics, such as O:C, H:C and average molecular formula, calculated using the NTA methodology which

indicated the OA to be predominantly comprised of CHO compounds, on average contributing 90 % to the total detected mass.

The NTA molecular formula assignments were used to investigate the composition of domestic BBOA derived from domestic

BB emissions via carbon number, double bond equivalents (DBE) (Eq. 1) and aromaticity index (AI) distributions (Koch and

Dittmar, 2006).

$$DBE = 1 + C - \frac{H}{2} + \frac{N}{2} \tag{1}$$

$$DBE_{\mathrm{AI}} = 1 + C - \frac{O}{2} - S - \frac{H}{2} - \frac{N}{2} \tag{2}$$

$$C_{\mathrm{AI}} = C - \frac{O}{2} - S - N \tag{3}$$

$$AI = \frac{DBE_{\mathrm{AI}}}{C_{\mathrm{AI}}} \tag{4}$$





### 3.2.1 Chemical composition of organic aerosol derived from flaming dominated emissions

In POA from flaming emissions, CHO compounds have two main regions of high abundance between $C_8$-$C_{11}$ and between $C_{13}$-$C_{17}$ as shown in Fig. 2. In the first region the DBE ranges between 4-7 (Fig. 2) which is indicative of aromatic species which typically possess a DBE of 4 or more. The presence of DBE values greater than 4 coupled with >$C_6$ suggests these

CHO species could be functionalised monoaromatics or small oxygenated polyaromatic species, for instance, napthalene-like species ($C_{10}$) which comprise two fused aromatic rings. Using the aromaticity index (Koch and Dittmar, 2006) (Eq.2-4) to classify species as non-aromatic, monoaromatic or polyaromatic, 51 % and 6% of the detected CHO mass was monoaromatic and polyaromatic respectively. Between $C_8$-$C_{11}$ the AI suggests approximately 42% of the mass in this region as monoaromatic in nature (Fig.A3). This coupled with > $C_6$ strongly indicates the presence of functionalised monoaromatics in the first region,

such as, phenoxyacetic acid ($C_8H_8O_3$) and phenyl acetic acid ($C_8H_8O_2$) which were identified using authentic standards. In the second region of high abundance between $C_{13}$-$C_{17}$ the DBE has a larger range of 6-12 (Fig. 2). From Fig. 2 and Fig. A3, the second region at $C_{13}$-$C_{17}$ contains DBE values which are generally double that of the first region and the AI suggests that the mass in this region is predominantly monoaromatic in nature (65 %). This suggests these compounds contain two aromatic rings linked via short sections of C-H and C=O bonds reflecting the structure of lignin. Figure A3 also shows a small

contribution of polyaromatic compounds in the $C_{13}$-$C_{17}$ region, with a relative contribution of 10 % on average. This is in accordance with observations of PAH emissions in previous studies from flaming combustion (Sekimoto et al., 2018; Stefenelli et al., 2019; Bertrand et al., 2018), however, it is clear for this study monoaromatics are of greater quantity in fresh emissions. In a NTA of ambient BBOA, Brege et al. (2018) observed a peak in relative abundance of CHO species at $C_{15}$ - $C_{16}$ which was attributed to terpene SOA products. However, in Fig. 2 there is no evidence of sesquiterpene ($C_{15}H_{24}$) derived SOA products

which would have relatively low DBE values. Liang et al. (2022) previously observed chamber studies often underrepresent BB terpene sources due to the lack of distillation from nearby heated and unburnt vegetation. Given that domestic BBOA is the combustion of non-living material, terpene derived SOA products could be more important in ambient BBOA from wildfires or crop burning in the presence of live vegetation.

     After photo-oxidation inside the chamber, the CHO carbon distribution is shifted to lower carbon number species ($C_7$-$C_{10}$)

indicating the fragmentation of the larger species with aging. At the same time the main peak in the oxygen distribution increases from $C_xH_yO_2$ to $C_xH_yO_4$ indicating more oxidised CHO compounds in the aged aerosol (Fig. A4). Li et al. (2021) suggested the higher $NO_x$ concentrations from flaming emissions could promote fragmentation pathways through the reactions of peroxy radicals ($RO_2$ + $HO_2$) with NO. However, other processes such as photolysis and heterogeneous photo-oxidation may also result in the production of small molecules. The aged $C_7$-$C_{10}$ CHO compounds possess DBEs in the range of 2-6

indicating some formation of non-aromatic compounds. However, using the AI, after atmospheric aging the CHO composition of the OA from flaming dominated emissions still contained a large degree of aromatic character (41%). The largest peak in the aged distributions in Fig. 2 at $C_8$ has a DBE value of 6 and is predominantly monoaromatic in nature (Fig. A3). The mass of this peak is dominated by $C_8H_6O_4$ which can be attributed to phthalic acid from previous observations (Wang et al., 2017a; Qi et al., 2019) and $C_8H_8O_3$ was confirmed as vanillin using an authentic standard.



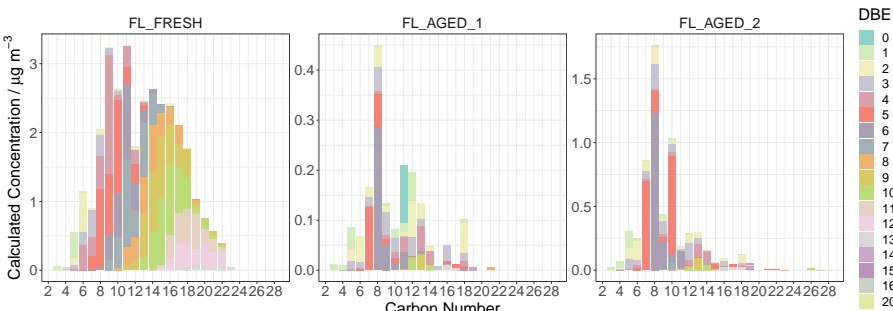

**Figure 2.** Carbon number vs concentration distribution coloured by the Double Bond Equivalent (DBE) of CHO compounds present in the POA from fresh emissions from flaming dominated burning conditions and after aging (POA+oPOA+SOA). The different filter sample IDs from the flaming dominated combustion experiments are given in each panel

For the CHON species, POA from flaming emissions has a main peak in the carbon number distribution at $C_6$ (Fig. A5) and approximately 74 % of the detected CHON mass in fresh OA has an O:N ratio $\geq$ 3 suggesting the presence of a nitro (-$NO_2$) group. Using a modified AI calculation derived for this work to account for the presence of the two N-O bonds within a -$NO_2$ group (Eq 5-7), 42% of the detected CHON mass is aromatic in nature. As shown in Fig. A5 the aromatic mass is predominantly comprised of $C_6$ monoaromatic compounds which coupled with the large proportion of nitro containing

compounds is highly indicative of nitro-monoaromatic species. For instance, 3-nitrophenol ($C_6H_5NO_3$) and 4-nitrocatechol ($C_6H_5NO_4$) were detected in the fresh OA and have been commonly observed as tracers of BB in previous studies (Claeys et al., 2012; Kourtchev et al., 2016; Iinuma et al., 2010; Kitanovski et al., 2012; Budisulistiorini et al., 2017; Li et al., 2017). After photo-oxidation the relative ratio of CHO:CHON concentration decreases from 41.5 to 8.7-9.2 (Table A1) indicating an increased contribution of CHON compounds to the aged OA composition. In the aged OA, the abundance of larger compounds

(ie. $> C_{10}$) increases, in particular polyaromatic $C_{12}$-$C_{14}$ species, such as $C_{12}H_9NO_4$ which accounts for 10% of the total CHON aromatic mass (Fig. A4). Zhang et al. (2013) previously observed $C_{12}H_9NO_4$ in ambient OA from the Los Angeles basin which had significant contributions from anthropogenic emissions and wood burning sources, however it was attributed to a nitro-monoaromatic compound. Whereas, in this work we assign $C_{12}H_9NO_4$ as a derivative of napthalene using the modified AI in Eq 5-7. Furthermore, the percentage mass of CHON with an O:N ratio $\geq$ 3 remained largely unchanged ($\approx$ 70 %) after aging

indicating CHON in aged OA are also predominantly NACs. Overall, the aged OA CHON composition contains a similar contribution of NACs to the POA which could be a result of the combination of unreacted species, loss of oxidation products to the gas phase and the condensation of new secondary products to the particle phase.





$$DBE_{\mathrm{AI}} = 1 + C - \frac{O-2}{2} - S - \frac{H}{2} - \frac{N-1}{2} \tag{5}$$

$$C_{\mathrm{AI}} = C - \frac{O-2}{2} - S - (N-1) \tag{6}$$

$$AI = \frac{DBE_{\mathrm{AI}}}{C_{\mathrm{AI}}} \tag{7}$$


### 3.2.2 Chemical composition of organic aerosol derived from smouldering dominated emissions

The measured carbon distribution of POA from a fresh smouldering dominated burn shows a peak between $C_8$-$C_{11}$ (Fig. 3) which largely has DBEs in the range of 4-8 and the AI estimated the majority of the mass in this carbon number region as aromatic (54.4%). The largest peak in the distribution shown in Fig. 3 is from $C_{10}$ compounds with a DBE of 6 predominantly

consisting of $C_{10}H_{10}O_3$ which was previously attributed to coniferylaldehyde in BBOA (Fleming et al., 2020; Smith et al., 2020). Furthermore, Figure A3 shows the majority of the CHO compounds are aromatic with a 50% and 16% contribution from monoaromatic and polyaromatic species respectively. This indicates the $C_8$-$C_{11}$ species are predominantly functionalised monoaromatic compounds as similarly observed for flaming. However, in this region there is also a greater concentration of polyaromatic compounds compared to flaming OA (see Figure A3), such as $C_{11}H_8O_2$ and $C_{11}H_8O_3$ which are naphthoic acid

derivatives previously observed in primary and secondary wood combustion products (Bruns et al., 2015). From Figure A3 the smouldering dominant POA shows an increased concentration of smaller $C_{11}$-$C_{12}$ oxygenated PAHs (o-PAHs) compared to flaming dominant POA which has the largest contribution from $C_{14}$ o-PAHs. A previous study observed the formation of PAHs and o-PAHs was dependent on the temperature and oxygen availability observing at higher temperatures in the absence of oxygen larger PAHs can form whilst smaller PAHs arise at lower temperatures (Fitzpatrick et al., 2008). Therefore the

relative PAH concentration and PAH composition will be dependent on the burn phase. Furthermore, the observed increased contribution of o-PAHs in the smouldering dominated burn is in agreement to findings by Orasche et al. (2013). In addition to functionalised monoaromatic compounds contributing to the $C_8$-$C_{11}$ peak, such as, phenylacetic acid ($C_8H_8O_2$) and 3-(4-hydroxyphenyl)propionic acid ($C_9H_{10}O_3$), a previous study found smouldering had higher emissions of methoxyphenols (Kjällstrand and Olsson, 2004) which possess > $C_7$ and is consistent with the observed range of carbon numbers and DBE

values in Fig. 3.

After photo-oxidation, the main region in the carbon number distribution reduces to $C_5$-$C_8$ (Fig. 3) with a second prominent peak at $C_{10}$. The DBE range also decreases to 1-6 after aging. The AI values indicate a large reduction in aromaticity after atmospheric aging as the percentage contribution of aromatic CHO to the detected mass decreases from 66% to 13%. Brege et al. (2018) observed a comparable shift to lower DBEs (1-5) in ambient BBOA after aging and Fang et al. (2021) showed SOA

from oxidised smouldering POA had significant contributions from oxygenated aliphatic species. From the oxygen number distribution shown in Fig. A4 the main peak increases from $C_xH_yO_3$ to $C_xH_yO_4$ indicating the presence of more oxidised species in aged OA. Overall this suggests OH functionalisation products contribute to the aged OA as well as the significant fragmentation of aromatic ring containing species from the POA. In Fig. 3, the peaks between $C_4$-$C_6$ with DBEs of 1-2 are largely comprised of small but highly oxidised species such as $C_4H_8O_4$, $C_5H_{10}O_4$, $C_5H_8O_{3-5}$ and $C_6H_{10}O_{4-5}$. Kalogridis et al.



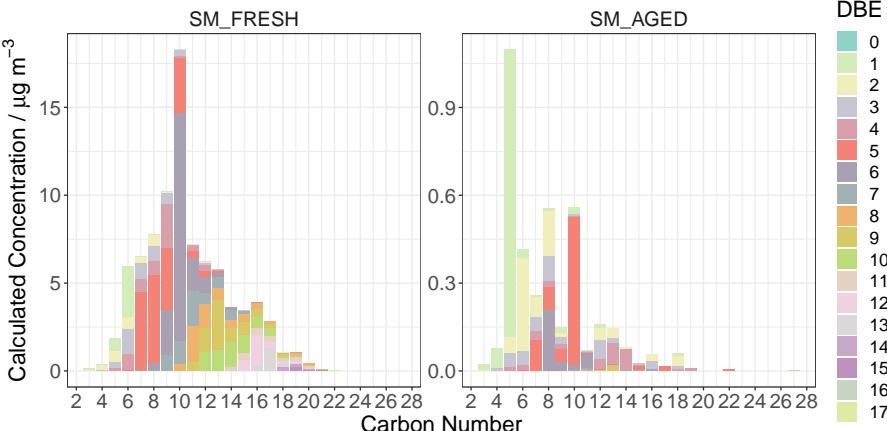

**Figure 3.** Carbon number vs concentration distribution coloured by the Double Bond Equivalent (DBE) of CHO compounds present in the POA from fresh emissions from smouldering dominated burning conditions and after aging (POA+oPOA+SOA). The different filter sample IDs from the smouldering dominated combustion experiments are given in each panel

(2018) observed higher emission factors of succinic and glutaric acids in smouldering compared to flaming therefore, these species could be derivatives of succinic acid ($C_4$) or glutaric acid ($C_5$) (Kalogridis et al., 2018; Kundu et al., 2010; Liang et al., 2021). This could also explain the lack of low DBE $C_4$ and $C_5$ compounds in the carbon number distribution derived from flaming dominated OA shown in Fig. 2. $C_5H_{10}O_4$ was also previously attributed to deoxyribose in BBOA (Smith et al., 2020), however this is likely not detected by the negative mode UHPLC-HRMS. The remaining aromatic mass after photo-oxidation

in smouldering dominated OA is predominantly formed of $C_7$-$C_8$ monoaromatic species (Fig. A3), such as, vanillin in addition to $C_8H_6O_4$ and $C_7H_6O_2$ which were previously attributed to phthalic acid and benzoic acid respectively (Wang et al., 2017a). Phthalic acid and benzoic acid were also identified as oxidation products of napthalene (Wang et al., 2017a) which is in agreement with the observed reduction in polyaromatic $C_{10}$-$C_{11}$ species in Fig. A3. Similar to flaming, polyaromatic species contributions were significantly reduced after aging in agreement with previous studies, which observed the emission factors

of o-PAHs to decrease by 20 % between fresh and aged BBOA (Li et al., 2020) and the degradation of particle bound PAHs after aging smoke particles (Kim et al., 2023b). Considering the damaging health impacts of oxygenated PAHs, the reduction of their contribution with aging could lead to important implications for the OA toxicity.

    For CHON species, the fresh OA distribution shows high concentrations at $C_9$ for nonaromatic compounds and $C_6$ for monoaromatic species (Fig. A5) which is similar to the peak in $C_6$-$C_{10}$ CHON species observed in ambient fresh BBOA

(Brege et al., 2018). These peaks predominantly consist of species such as $C_9H_{11}NO_4$ and $C_6H_5NO_5$ which were previously observed in ambient cloud water samples influenced by agricultural BB (Desyaterik et al., 2013) and in fresh BBOA from controlled burn experiments (Lin et al., 2016). In addition, 72% of the CHON mass had a O:N $\geq$ 3 suggesting the presence of -$NO_2$ functionality. This is therefore in accordance with Lin et al. (2017) who attributed $C_6H_5NO_5$ in BrC originating from a major BB event to nitrobenzenetriol. After photooxidation inside the chamber the CHO:CHON concentration ratio decreases





from 29.2 to 10.9 (Table A1) indicating a greater contribution of CHON species to the overall OA composition. Similar to flaming, the OA distribution in Fig. A5 shows an increase in larger CHON species (ie. > $C_{10}$) after aging. Monoaromatic compounds in the aged OA, are predominantly comprised of $C_5$, $C_6$ and $C_{12}$ species such as $C_5H_5NO_4$, $C_6H_5NO_3$, $C_6H_4N_2O_5$ and $C_{12}H_{12}N_2O_4$ (Fig. A5). $C_6H_5NO_3$ and $C_6H_4N_2O_5$ were identified as 3-nitrophenol and 2,4-dinitrophenol respectively using authentic standards and $C_5H_5NO_4$ was previously observed in BrC from a major BB event (Lin et al., 2017) but the structure could not be elucidated. However, a monoaromatic $C_5$ species is indicative of furanic origins as previous observations indicate furans are important for SOA production in smouldering fires (Stefenelli et al., 2019). In the aged OA, the relative contribution of aromatic compounds to the CHON mass decreased from 45% to 31 % and the proportion of compounds with O:N $\geq$ 3 reduced to 47% which overall indicates a reduction in the contribution of NACs to the OA composition after atmospheric aging.

## 3.3 Impact of the burn phase on the oxidation and aged chemical composition of organic aerosol

As discussed in Sect. 3.2.1 and 3.2.2 the burning conditions influence the POA composition and POA mass with subsequent atmospheric aging producing two unique distributions (see Fig. 2 and Fig. 3), which could enable the development of burn-specific tracer species. However, overall there is a comparable contribution of aromatic species (> 50%) to the POA composition under both conditions which is in accordance with Akherati et al. (2020) and Gilman et al. (2015) who observed oxygenated aromatics had the greatest SOA formation potentials and contributed to nearly 60 % of the SOA mass from BB. However after aging, the change in the contribution of aromatic compounds to the OA composition differed with a significant reduction observed for smouldering dominated compared to flaming dominated burns. In particular, smouldering dominated emissions showed a greater reduction in polyaromatic contributions to the OA mass after aging (-16.2%) compared to the flaming dominated phase (-2.6%). This difference in compositional change indicates aging of OA from smouldering emissions could result in more oxidised products compared to flaming.

In order to compare the effect of atmospheric aging processes on the OA composition from domestic BBOA, O:C ratios of aromatic and non-aromatic compounds were examined via their probability density distributions which visualise the differences in the population of observable O:C values. The distribution shown in Fig. 4 is constructed using a kernel density estimation which fits a smooth distribution across a series of band widths to a histogram of the observed O:C values in the domestic BBOA samples. The *y axis* represents density meaning the probability of the OA having a certain O:C value can be computed by integrating the area under the curve. In this analysis the peaks in the O:C distributions of fresh and aged OA are examined to observe differences in the composition and hence provide insight into the magnitude of OA oxidation under different burning conditions. Generally, the peak of the O:C distributions in Fig. 4 of aromatic compounds for flaming dominated and smouldering dominated experiments show a greater change between fresh and aged aerosol compared to non-aromatic species. This demonstrates that oxidation of aromatic compounds is significant for the observed compositional change as inferred from Sect. 3.2.1 and 3.2.2.

In the POA from fresh emissions, the average O:C for CHO compounds is generally higher for the smouldering dominated phase than the flaming dominated phase(see Table A1). The smouldering dominated POA shows a broader distribution in in Fig.





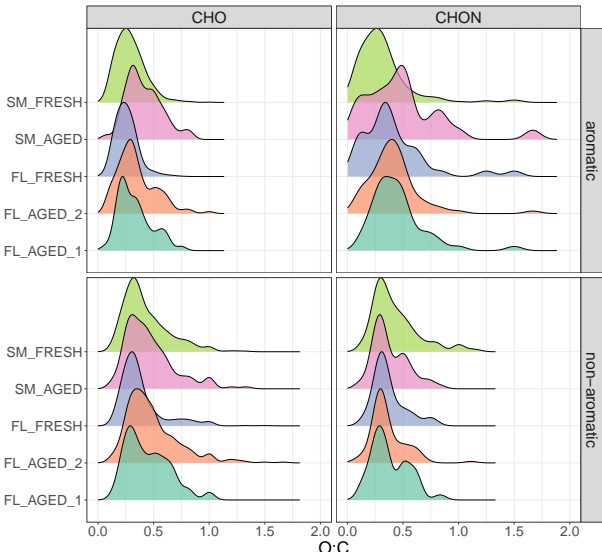

**Figure 4.** Probability density distribution of O:C ratios of the detected aromatic (top panel) and non-aromatic (bottom panel) CHO (left panel) and CHON (right panel) compounds. The *y axis* height represents density and the area under the curve represents probability. Sample IDs for each burn experiment are given on the *y axis* and the distribution colour.

4 indicating the presence of more oxygenated compounds than from the flaming dominated burns, as seen in the oxygen number
distribution (Fig. A4). After aging, the distributions shift to higher O:C values consistent with the observed increased oxygen content and fragmentation of the carbon backbone indicative of the oxidation of OA (Jimenez et al., 2009). For aromatic CHO compounds the increase in O:C after aging is greater in the smouldering dominated burn compared to the flaming dominated burns. In smouldering dominated OA the main peak in the O:C ratio for aromatic CHO compounds increased from 0.25 to 0.50 with a smaller peak at 0.80. Upon aging in the flaming dominated burn the main peak of the O:C ratio increased from 0.22 to
0.29 with a smaller peak at 0.50. Furthermore, the area under the distribution represents probability and at the O:C value of 0.50, the area was greater for smouldering dominated than flaming dominated phases suggesting a larger number of oxidised species. Additionally the presence of high O:C values ($\approx$ 0.80) from the smouldering dominated phase derived OA which are absent in the flaming dominated burn distribution similarly indicates a greater proportion of increasingly oxidised compounds. This is in agreement with Li et al. (2021) who observed greater oxidation of smouldering emissions compared to flaming.
For aromatic CHON species, there is a significant difference in the POA probability density distributions with a broader distribution in the smouldering dominated phase compared to 3 resolved peaks for the flaming dominated phase. The smouldering dominated POA distribution peaks at an O:C value of 0.25 whereas the flaming dominated POA distribution contains 3 distinct peaks at O:C ratios of 0.15, 0.34 and 0.60 with the greatest density at 0.34. However, the lower value of the O:C ratio peak in smouldering dominated POA compared to flaming dominated POA could be due to the greater contribution of polyaromatic





species (Fig. A3) which typically possess relatively low O:C values. Similarly, after aging the O:C distributions in Fig. 4 show different trends for flaming dominated and smouldering dominated experiments. In smouldering dominated aged OA there is a clear increase in O:C from 0.25 to 0.47 which is in agreement with previous observations of the change in O:C (0.09-0.30) from smouldering fires (Bertrand et al., 2017; Grieshop et al., 2009; Tasoglou et al., 2017). Whereas for flaming, the distributions of aromatic CHON compounds are relatively similar between fresh and aged OA and the increase in O:C from the main peak

is relatively low (0.07). $C_6H_5NO_3$ (nitrophenol), $C_7H_7NO_4$ (nitroguaiacol) and $C_{10}H_7NO_3$ (nitro-1-napthol) were identified in both POA and aged OA from the flaming dominated phase with an increase in concentration after aging, indicating the formation of similar CHON compounds to POA during photo-oxidation.

Overall, the observed values of O:C in CHO compounds for both flaming and smouldering in Fig. 4 are in a similar range to O:C reported in ambient BBOA (0.42 - 0.47) (Dzepina et al., 2015; Brege et al., 2018). Furthermore, CHON species from

ambient OA influenced by varying degrees of BB were reported to have O:C values in the range of 0.37-0.50 (Lin et al., 2012; Kourtchev et al., 2016; An et al., 2019; Dzepina et al., 2015) which is similar to the peak O:C range shown in Fig. 4. However, the O:C values reported in these studies include both aromatic and non-aromatic species and are weighted by relative abundance derived from limited metrics, such as, peak area or the number of detected formulas.

Van Krevelen diagrams of the O:C vs H:C space can also provide insight into the differences in composition and atmospheric

aging between smouldering and flaming derived OA (Fig. 5). For instance, POA derived from fresh smouldering dominated emissions has a greater range of O:C values across a similar range of H:C ratios as fresh flaming dominated emissions in Fig. 5 suggesting increasingly oxygenated OA. In addition, Fig. 5 shows POA from fresh smouldering dominated emissions has a greater quantity of polyaromatic CHO and CHON compounds as previously observed in Fig. A3 and Fig. A5. After aging, there is a significant reduction in aromatic species for the smouldering dominated phase and to a lesser extent in the flaming

dominated phase (Fig. 5). In the aged OA from smouldering dominated emissions the reduction in polyaromatic compounds is significantly greater compared to flaming dominated derived OA, with almost complete loss of the polyaromatic CHO species in Fig. 5. Whereas for flaming dominated OA, the number of aromatic CHON species in the Van Krevelen space increased after aging, notably for polyaromatic CHON species. This trend can also be seen for the flaming experiments in Fig. A5 with increased $C_{12}$-$C_{14}$ polyaromatic CHON species in aged OA and a simultaneous decrease of $C_{12}$-$C_{14}$ polyaromatic CHO species

in Fig. A3. Therefore, this suggests the formation of NACs from the oxidation of aromatic CHO compounds during the flaming dominated phase. The same trend was also observed in the Van Krevelen of species detected in positive mode ESI (Fig. A6). The production of NACs could be greater from the flaming dominated burn since $NO_x$ emissions from flaming are typically higher than smouldering (Fig. A1a) thus leading to the formation of ring retained nitroaromatic species.

The notable variation of the aromatic contribution to aged OA compositions and significant differences in the concentration

of polyaromatic species between burn phase could thereby have important implications for toxicity. Kim et al. (2023b) previously investigated the toxicity of wood smoke particles and observed that after aging the mutagenicity was lower compared to fresh particles when considering PAHs. Furthermore, on an equal particle mass basis aged flaming smoke particles had a higher potential toxicity value with respect to PAHs than aged smouldering smoke particles (Kim et al., 2023b). This could potentially be the result of an increased fraction of nitro-PAH compounds from flaming (Fig. 5), some of which are known to





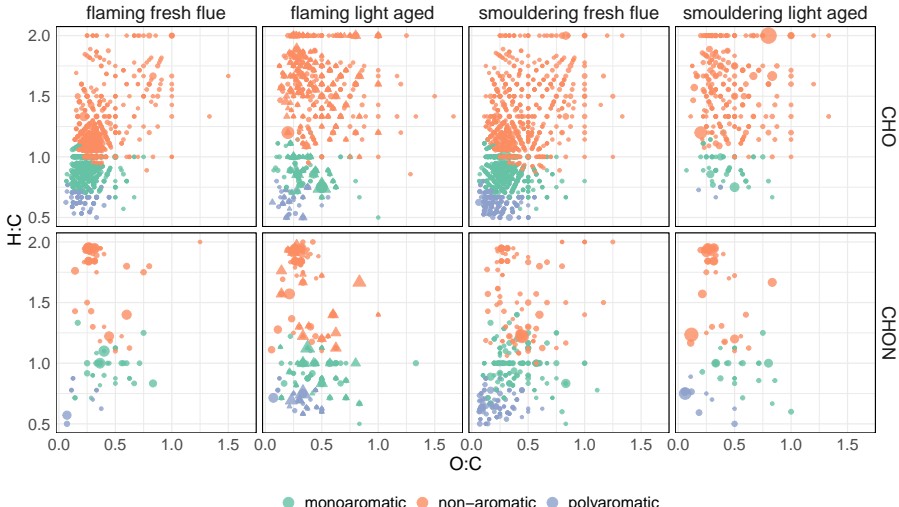

**Figure 5.** Van Krevelen diagrams of H:C vs O:C ratios for CHO (top row) and CHON (bottom row) compounds derived from flaming dominated and smouldering dominated burn phases, coloured by the aromaticity index assignment (non-aromatic: AI < 0.5, monoaromatic: $0.5 \leq$ AI $\leq 0.67$, polyaromatic: AI > 0.67). In the flaming light aged panel the point shape represents the two repeats: FL_AGED_1 (circles) and FL_AGED_2 (triangles)

exceed the toxicity of their parent PAH. Therefore, the OA composition under different burning conditions can be an important factor affecting the toxicity but often the greater observed emission factors from smouldering (Zhang et al., 2022; Jen et al., 2019) prevail in the determination of the toxicity.

    The volatility of the domestic BBOA was estimated using the 2-dimenstional volatility basis set (2D-VBS) parameterisation described in Li et al. (2016) (see Appendix A1) to determine differences in aging processes which could explain the observed

differences in composition. The 2D-VBS estimation error was shown to increase for lower saturation mass coefficients therefore some caution must be taken when considering low volatility products (Li et al., 2016). Figure A7 shows the Van Krevelen distribution coloured by the OA volatility which indicates after aging there is a slight increase in volatility likely from the formation of smaller compounds as shown in Sect. 3.2.1 and 3.2.2. For the smouldering dominated burn aging resulted in the almost complete loss of low volatile organic compounds (LVOC) and semi-volatile organic compounds (SVOC) from the aged

OA. OA from the flaming dominated phase in the region of H:C < 1 and O:C < 0.5 conversely showed a notable increase in the SVOC CHON compounds and a simultaneous reduction of SVOC CHO compounds, suggesting the formation of NACs after aging as observed in Fig. 5. This difference in processing between the burning conditions is in accordance with Kalogridis et al. (2018) who similarly observed changes in OC from flaming to be driven by the production or partitioning of organic compounds to the particle phase whereas for smouldering, evaporation of semi-volatile species was considered an important

sink of OC.





Generally, the observed O:C ratios of the aromatic compounds in aged OA shown in Fig. 5 are in agreement with those for SOA derived from aromatic oxidation (eg. toluene, m-xylene and napthalene) with reported values ranging between 0.57-0.75 (Chen et al., 2021, 2020; Loza et al., 2012; Chhabra et al., 2010) and the observed range for phenolic SOA oxidation products (0.3-1.0) (Ofner et al., 2011). Furthermore, the observable H:C vs O:C space in Fig. 5 is within the range reported

for lignin-like compounds (1-1.5 vs 0.2-0.6) (An et al., 2019) as expected for BBOA. Overall, these results show SOA from aromatic compounds via ring-opening and ring retained nitroaromatic formation routes contribute substantially to the aged OA composition and may have important implications for atmospheric chemistry (Bloss et al., 2005).

## 4    Conclusions

The chemical composition of domestic biomass burning organic aerosol (BBOA), from a series of controlled burn chamber

experiments, was investigated using a newly developed semi-quantitative non-target analysis (NTA) methodology to understand the bulk compositional changes occurring from atmospheric aging under different burning conditions (ie. flaming dominated and smouldering dominated phases). Overall, the composition of domestic BBOA was dominated by oxygenated compounds (CHO), on average contributing to 90% of the total detected mass with a smaller contribution ( < 10%) of organonitrogen species (CHON). The NTA approach enabled significant compositional differences between the OA derived from emission

dominated by flaming or smouldering phases to be observed. Firstly, the estimated concentrations of the detected compounds in the OA were markedly higher from the smouldering dominated emissions compared to flaming dominated emissions, as a result of the larger OA emission factors associated with smouldering. This indicates the burn phase in a domestic environment is a critical factor for controlling indoor air pollutant concentrations along with air filtration and the stove model (Ward et al., 2015). Considering the OA chemical composition from fresh emissions, flaming dominated POA had a large contribution of CHO

compounds between $C_8$-$C_{11}$ and $C_{13}$-$C_{17}$ which were predominantly comprised of functionalised monoaromatic compounds. Smouldering dominated POA emissions had a higher concentration of lower molecular weight CHO species, predominantly in the region of $C_8$-$C_{11}$ with a peak at $C_{10}$, which were also attributed to functionalised monoaromatic compounds. Furthermore, smouldering dominated POA also had a greater percentage contribution from o-PAHs of 16% over the same carbon range compared to the flaming dominated POA (6%) which has important implications for the toxicity of POA. For CHON species,

the observed POA composition contained comparable concentrations of $C_6$ monoaromatic species between the burn phases that were largely assigned as NACs such as widely used BB tracers, nitrophenols and nitrocatechols. However, after aging, the OA composition between the burning conditions significantly diverged particularly in the relation to the contribution of aromatic CHO and CHON compounds to the aged OA composition which was attributed to burn specific aging processes. For OA from the smouldering dominated phase, aging decreased the relative contribution of aromatics with almost complete reduction of

both polyaromatic CHO and CHON species resulting in the formation of ring-opened products. In comparison, for the flaming dominated burns, the reduction in the contribution of aromatic compounds to the detected OA mass after aging was less than in smouldering and Van Krevelen analysis indicated the number of polyaromatic CHON species notably increased, suggesting the formation of NACs from aromatic CHO species. These differences in the aromatic contribution to the OA composition between



the burn phases have important implications for toxicity, particularly in relation to polyaromatic species which are known carcinogenic species. The formation of ring retained NACs from the flaming dominated burns highlight important implications for both toxicity and BrC formation. In contrast, higher volatility ring-opened products were an important contribution to OA from aged smouldering dominated emissions. These products could volatilise from the particulate phase and impact on atmospheric chemistry and $O_3$ formation, invariably leading to the creation of compounds of unknown toxicities. At present, toxicology endpoints used for policy-making decisions on mitigating impacts for human health are typically based on mass. In reality, this is a more complex picture with multiple factors affecting the toxicity of domestic BBOA such as the emission factor of a compound, the OA composition as studied here, the total mass of fuel burnt and ultimately the length of time exposed to the emission.

*Author contributions.* RLE prepared the manuscript with contributions from co-authors. The wood burning experiments were predominantly designed by AV and GM with input from all authors. Chamber experiments were performed by AV, DH, HW, SAS, OEO and RLE. SAS and HW provided trace gas and AMS measurements from the Manchester Aerosol Chamber. Offline filter sample measurements and non-target analysis was conducted by RLE. DJB contributed to scientific discussion. JFH and ARR supervised the project and obtained funding to develop the methodology.

*Competing interests.* The authors declare no competing interests

*Acknowledgements.* The Manchester Aerosol Chamber receives funding from the Horizon 2020-Research and Innovation Framework Programme, H2020-INFRAIA-2020-1, Sustainable Access to Atmospheric Research Facilities (ATMO-ACCESS), Grant Agreement number: 101008004. The Orbitrap MS at the University of York was funded by a Natural Environment Research Council (NERC) strategic capital grant (grant no. CC090). The authors thank the NERC Panorama Doctoral Training Partnership (DTP), under grant NE/S007458/1 for the studentship of Rhianna Evans. Sara Syafira acknowledges studentship support from the Indonesia Endowment for Education (LPDP) and Osayomwanbor Oghama acknowledges studentship support from the Tertiary Education Trust Fund (TETFund), Nigeria. Daniel Bryant acknowledges financial support from NERC under grant NE/W002051/1 and NE/S010467/1.





## Appendix A

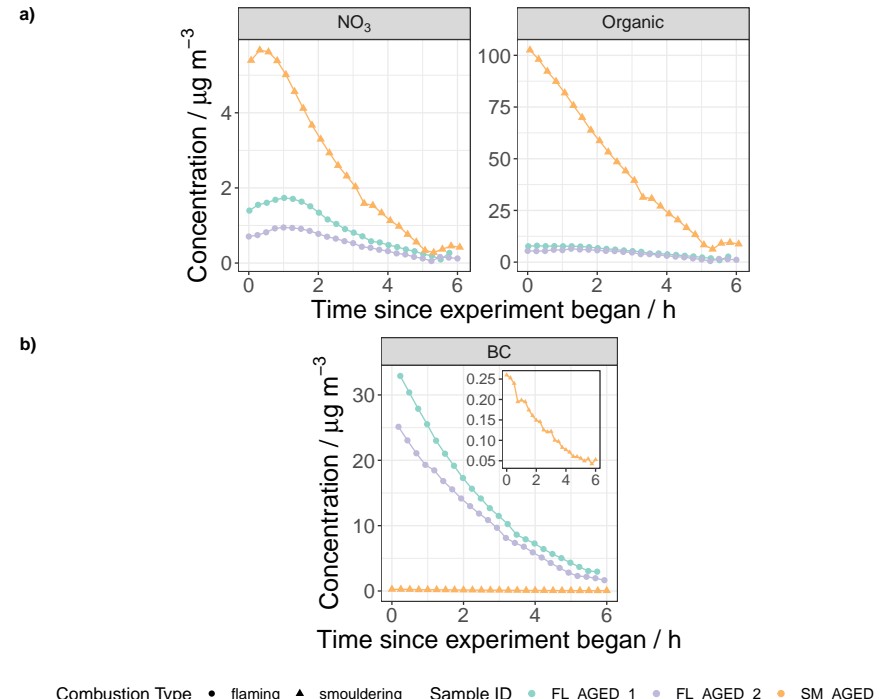

**Figure A1.** a) Online AMS measurements of NO$_3$ and organic carbon concentrations for flaming dominated (circle points) and smouldering dominated (square points) aging experiments, averaged to 15 minute intervals, b) BC timeseries averaged to 5 minutes, with sample IDs given by the point or line colour.





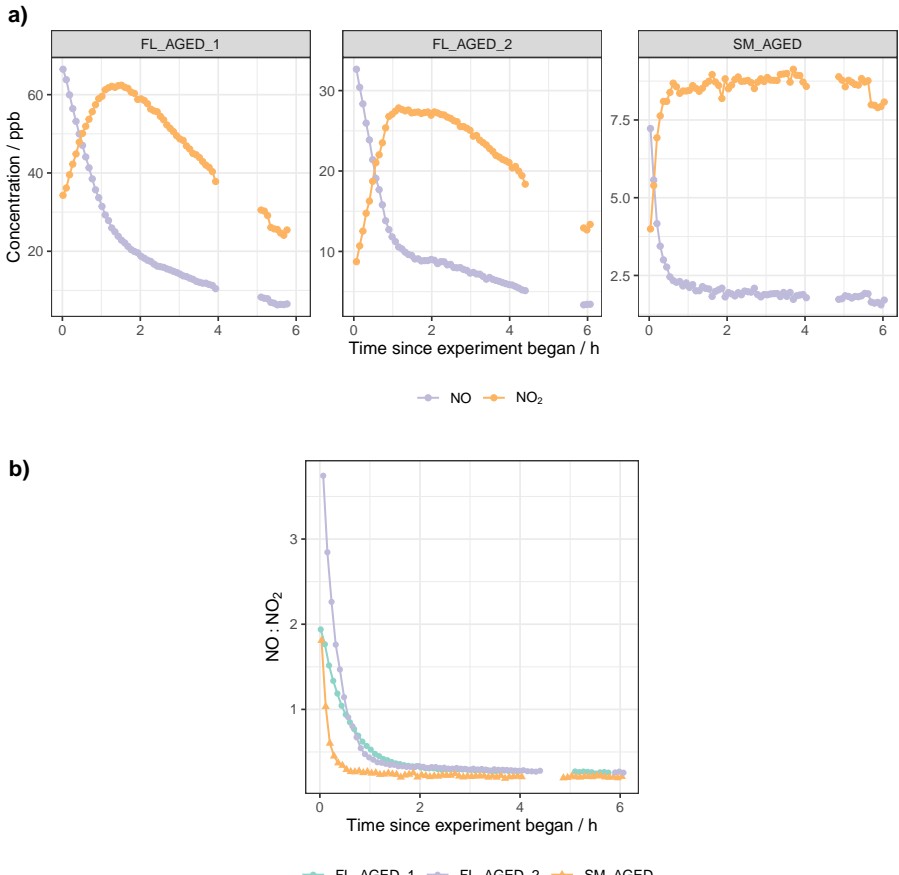

**Figure A2.** a) NO and $NO_2$ timeseries of the flaming dominated and smouldering dominated aging experiments with individual experiments shown in each panel. b) Ratio of $NO:NO_2$ during the flaming dominated and smouldering dominated aging experiments with the experiment indicated by the line colour. In panel a) and b) data is removed during filter sampling due to interference with the $NO_x$ instrument inlet.





**Table A1.** Average aerosol metrics calculated from the semi-quantitative NTA methodology of the detected CHO and CHON compounds in the domestic BBOA samples

| Sample ID | Category | O:C | H:C | Molecular formula | Relative abundance / % | DBE |
|-----------|----------|-----|-----|-------------------|------------------------|-----|
| FL_FRESH | CHO | 0.28 | 1.05 | $C_{12.8}H_{13.0}O_{3.3}$ | 97.2 | 7.32 |
| FL_AGED_1 | CHO | 0.42 | 1.39 | $C_{10.1}H_{14.6}O_{3.8}$ | 84.6 | 3.82 |
| FL_AGED_2 | CHO | 0.41 | 1.19 | $C_{9.3}H_{11.3}O_{3.5}$ | 83.6 | 4.68 |
| SM_FRESH | CHO | 0.35 | 1.04 | $C_{10.6}H_{10.4}O_{3.3}$ | 96.1 | 6.41 |
| SM_AGED | CHO | 0.54 | 1.56 | $C_{8.0}H_{12.1}O_{3.7}$ | 84.9 | 2.95 |
| FL_FRESH | CHON | 0.54 | 1.74 | $C_{13.5}H_{23.1}O_{5.2}N_{1.6}$ | 2.3 | 3.76 |
| FL_AGED_1 | CHON | 0.41 | 1.25 | $C_{11.2}H_{14.5}O_{4.3}N_{1.5}$ | 9.7 | 5.62 |
| FL_AGED_2 | CHON | 0.4 | 1.61 | $C_{14.2}H_{23.3}O_{4.6}N_{1.6}$ | 9.1 | 4.38 |
| SM_FRESH | CHON | 0.42 | 1.18 | $C_{10.4}H_{12.3}O_{3.7}N_{1.3}$ | 3.3 | 5.91 |
| SM_AGED | CHON | 0.35 | 1.48 | $C_{14.7}H_{21.7}O_{4.1}N_{2.1}$ | 7.8 | 5.91 |



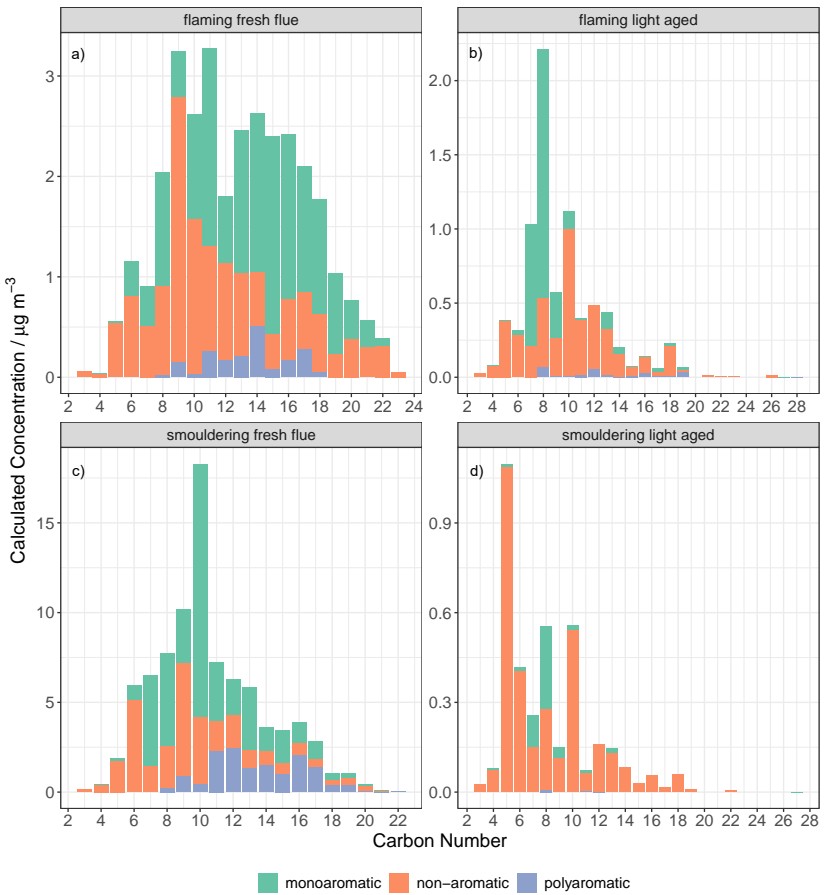

**Figure A3.** Carbon number distribution of CHO compounds present in the fresh flue emissions (POA) and light aged aerosol (POA+oPOA+SOA) from flaming dominated (a-b) and smouldering dominated (c-d) combustion. The two repeats of the flaming light aged experiment were combined to produce a total concentration in the figure. Coloured by the aromaticity index assignments of non-aromatic (AI < 0.5), mono-aromatic ( $0.5 \leq$ AI $\leq 0.67$) and polaromatic (AI > 0.67))

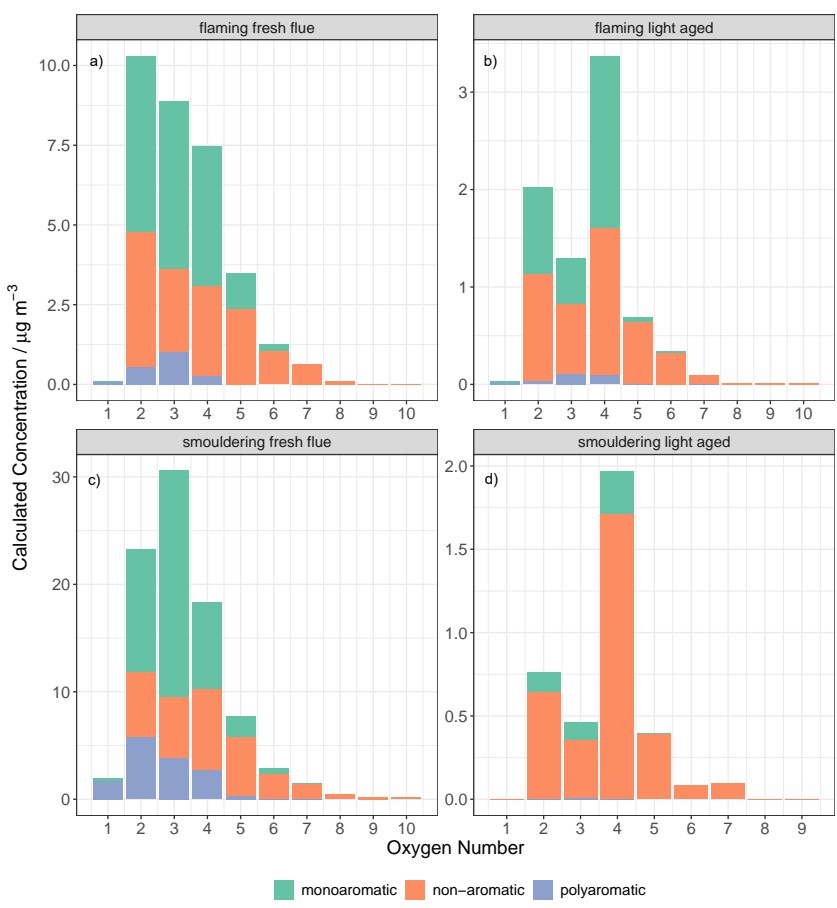

**Figure A4.** Oxygen number distribution of CHO compounds present in the fresh flue emissions (POA) and light aged aerosol (POA+oPOA+SOA) from flaming dominated (a-b) and smouldering dominated (c-d) combustion. The two repeats of the flaming light aged experiment were combined to produce a total concentration in the figure. Coloured by the aromaticity index assignments of non-aromatic (AI < 0.5), mono-aromatic ( $0.5 \leq$ AI $\leq 0.67$) and polaromatic (AI > 0.67)





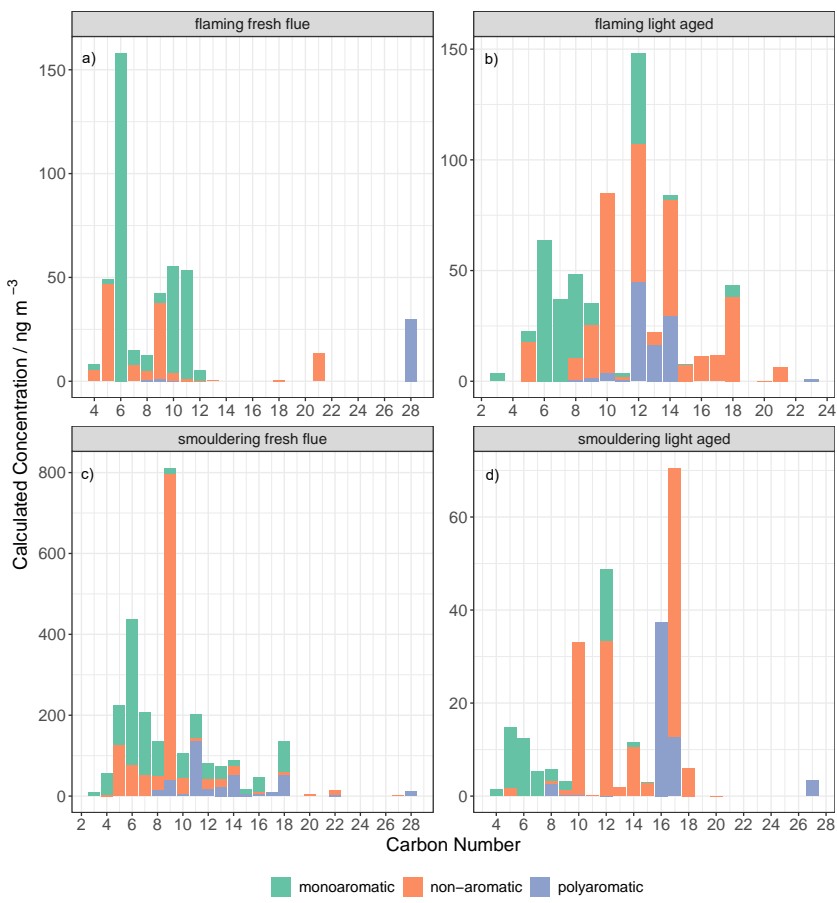

**Figure A5.** Carbon number distribution of CHON compounds present in the fresh flue emissions (POA) and light aged aerosol (POA+oPOA+SOA) from flaming dominated (a-b) and smouldering dominated (c-d) combustion. The two repeats of the flaming light aged experiment were combined to produce a total concentration in the figure. Coloured by the aromaticity index assignments of non-aromatic (AI < 0.5), mono-aromatic ( $0.5 \leq$ AI $\leq 0.67$) and polaromatic (AI > 0.67)



**Volatility Parameterisation** The volatility of the organic aerosol was calculated according to the parameterisation described in Li et al. (2016) using Eq. A1:

$$\log_{10} C_0 = (c^0 - c) \times b_C - n_O \times b_O - 2\frac{n_C \times n_C}{n_C + n_O} \times b_{CO} - n_N \times b_N - n_S \times b_S \tag{A1}$$

where $c^0$ is the reference carbon number, $n_C$, $n_O$, $n_N$, $n_S$ represent the number of carbon or oxygen or nitrogen or sulfur atoms present in the structure, $b_C$, $b_O$, $b_N$, $b_S$ represents the atom contribution to $\log_{10} C_0$ and $b_{CO}$ is the carbon-oxygen non-ideality.

Compounds were classed as volatile organic compounds (VOC) when $C_0 > 3 \times 10^6$ $\mu$g m$^{-3}$, intermediate volatile organic compounds (IVOC) when $300 < C_0 < 3 \times 10^6$ $\mu$g m$^{-3}$, semi-volatile organic compounds (SVOC) when $0.3 < C_0 < 300$ $\mu$g m$^{-3}$, low volatile organic compounds (LVOC) when $3 \times 10^{-4} < C_0 < 0.3$ $\mu$g m$^{-3}$ and extremely low volatile organic compounds

(ELVOC) when $C_0 < 3 \times 10^{-4}$ $\mu$g m$^{-3}$.



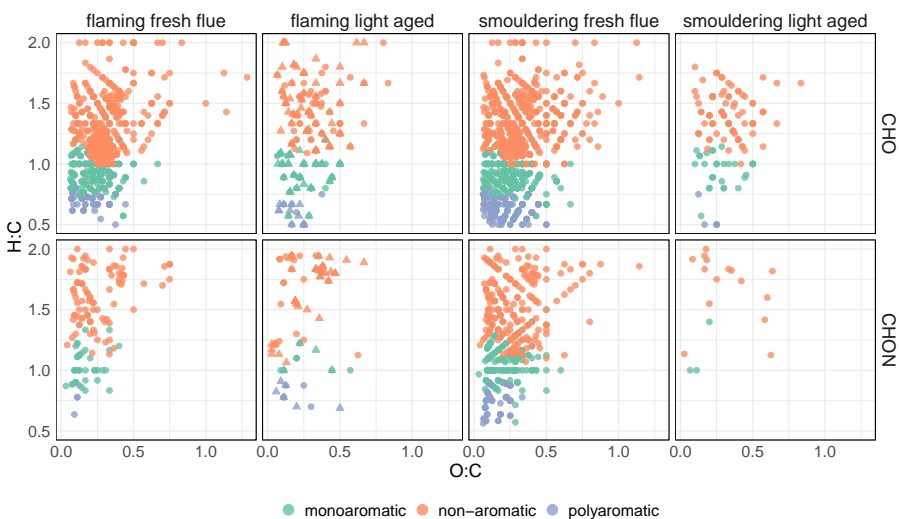

**Figure A6.** Van Krevelen diagrams of H:C and O:C ratios for CHO (top row) and CHON (bottom row) compounds in OA derived from flaming dominated and smouldering dominated burn phases and detected by positive mode ESI. aromaticity index (AI) assignments are shown by the point colour (non-aromatic: AI < 0.5, monoaromatic: $0.5 \leq$ AI $\leq 0.67$, polyaromatic: AI > 0.67). In the flaming light aged panel the point shape represents the two flaming repeats: FL_AGED_1 (circles) and FL_AGED_2 (triangles)





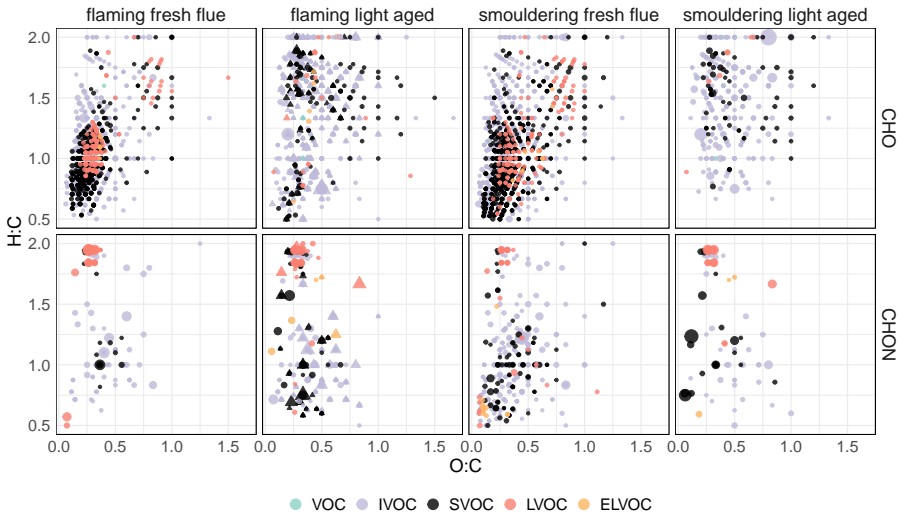

**Figure A7.** Van Krevelen diagrams of H:C and O:C ratios for CHO (top row) and CHON (bottom row) compounds in the OA derived from flaming dominated and smouldering dominated burn phases. In the flaming light aged panel the point shape represents the two repeats: FL_AGED_1 (circles) and FL_AGED_2 (triangles). Compounds are coloured by the volatility estimations described in the Li et al. (2016) parameterisation for volatile organic compounds (VOC), intermediate volatile organic compounds (IVOC), semi-volatile organic compounds (SVOC), low volatile organic compounds (LVOC) and extremely low volatile organic compounds (ELVOC)



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
