# Peer review of "The importance of burning conditions on the composition of domestic biomass burning organic aerosol and the impact of atmospheric aging"

_EGUsphere, 2024_

## Author Comment (AC1)

**Reviewer 1**

*Evans et al. show the effects of flaming and smoldering biomass combustion on the emission chemical composition. Moreover, they also show differences after aging. Overall, the study is well-designed, and the paper is well-organized. The findings will benefit the community by helping them understand the effects of biomass-burning aerosols on climate. I have a few minor comments that I hope the author can consider.*

We thank the reviewer for taking the time to read our manuscript and providing insightful comments and suggestions. We hope the following explanations and clarifications are satisfactory in addressing the reviewers comments.

1.1 *It seems like the experiments have CO and CO2 measurements. If this is true, it will be better to quantify the combustion condition based on modified combustion efficiency (MCE), and I am interested to see the correlation between chemical composition and MCE.*

    The authors appreciate the reviewers suggestion and agree that MCE is widely used to distinguish flaming and smouldering therefore these values are now included in Table 1 instead of $CO:CO_2$. As the range of MCE studied is limited we did not correlate composition and MCE in our original manuscript. The identified intermediary burn also had a very similar MCE to the flaming experiments (0.95). Therefore, to do a correlation the authors would require more repeats across the full MCE range which was not possible in this campaign. Due to not measuring at the stove flue we cannot provide MCE measurements for the fresh flue filters.

Table 1: List of the OA samples used in this study and the initial conditions at the start of the aging period

| Experiment date | Conditions | Sample ID | Aging period / hrs | PM concentration / $\mu g\ m^{-3}$ | NO:NO$_2$ | OC:BC[*] | MCE |
|---|---|---|---|---|---|---|---|
| 21/04/2022 | Flaming light aged | FL_AGED_1 | 5:50 | 243.6 | 1.94 | 0.32 | 0.96 |
| 26/04/2022 | Smouldering light aged | SM_AGED | 6:05 | 213.6 | 1.81 | 406.3 | 0.78 |
| 28/04/2022 | Flaming light aged | FL_AGED_2 | 6:05 | 153.4 | 3.74 | 0.21 | 0.93 |
| 30/08/2022 | Flaming fresh flue | FL_FRESH | - | - | - | - | - |
| 31/08/2022 | Flaming fresh flue | SM_FRESH | - | - | - | - | - |

[*]total organic content measured by AMS

1.2 *For section 2.1.3, is there any reason why you don't use water:MeOh solution to extract the filter? If your samples were initially extracted by methanol, how would that affect water-soluble but methanol-insoluble species? And could you provide an estimation of how much organic will be lost during the process?*

    The primary reason for not using water to extract the filter is the potential production of OH radicals through the sonication step when using water ([1, 2]). Secondly, many studies have observed that the use of methanol increases the extraction efficiency for OC with studies reporting more than 90% extraction efficiency of OC from biomass burning $PM_{2.5}$ samples ([3, 4]). Therefore, we anticipate minimal loss of organic carbon in our extraction method. In addition, methanol extracts were found to be more light absorbing in previous studies due to the increased extraction of BrC components, which may otherwise be insoluble in water ([5–8]). Given biomass burning is a large source of BrC the use of methanol solvent is favourable in this respect.

1.3 *For Figure 1, I suggest adding a legend of markers as you did for other figures.*

    The authors appreciate this feedback and will add this to the manuscript. We have also added observations from aircraft campaigns and long term measurement sites.

[Figure]

**1.4 I think eq. 5-7 are duplicates of equ. 2-4.**

The authors clarify that the equations are not duplicates as for the aromaticity index calculation of CHON species we first subtract $NO_2$ for the nitro group from the formula and hence the O-2 and N-1 in Eq 5-7. However we will make this clearer by using alternate terminology in the equations.

"

$$DBE_{\text{CHON}} = 1 + C - \frac{O - 2}{2} - S - \frac{H}{2} - \frac{N - 1}{2} \tag{1}$$

$$C_{\text{CHON}} = C - \frac{O - 2}{2} - S - (N - 1) \tag{2}$$

$$AI_{\text{CHON}} = \frac{DBE_{\text{CHON}}}{C_{\text{CHON}}} \tag{3}$$

"

**Bibliography**

[1] B. Miljevic, F. Hedayat, S. Stevanovic, K. E. Fairfull-Smith, S. E. Bottle, and Z. D. Ristovski. "To Sonicate or Not to Sonicate PM Filters: Reactive Oxygen Species Generation Upon Ultrasonic Irradiation". In: *Aerosol Sci. Tech.* 48 (12 2014), pp. 1276–1284. DOI: 10.1080/02786826.2014.981330.

[2] K. Makino, M. M. Mossoba, and P. Riesz. "Chemical effects of ultrasound on aqueous solutions. Formation of hydroxyl radicals and hydrogen atoms". In: *J. Phys. Chem.* 87 (8 1983), pp. 1369–1377. DOI: 10.1021/j100231a020.

[3] Z. Xu, W. Feng, Y. Wang, H. Ye, Y. Wang, H. Liao, and M. Xie. "Potential underestimation of ambient brown carbon absorption based on the methanol extraction method and its impacts on source analysis". In: *Atmos. Chem. Phys* 22 (2022), pp. 13739–13752. DOI: 10.5194/acp-22-13739-2022.

[4] M. Xie, M. D. Hays, and A. L. Holder. "Light-absorbing organic carbon from prescribed and laboratory biomass burning and gasoline vehicle emissions". In: *Sci. Rep.* 7 (1 2017), pp. 1–9. DOI: 10.1038/s41598-017-06981-8.

[5] T. Cao, M. Li, C. Zou, X. Fan, J. Song, W. Jia, C. Yu, Z. Yu, and P. '. A. Peng. "Chemical composition, optical properties, and oxidative potential of water-and methanol-soluble organic compounds emitted from the combustion of biomass materials and coal". In: *Atmos. Chem. Phys* 21 (2021), pp. 13187–13205. DOI: 10.5194/acp-21-13187-2021.

[6] M. Xie, X. Peng, Y. Shang, L. Yang, Y. Zhang, Y. Wang, and H. Liao. "Collocated Measurements of Light-Absorbing Organic Carbon in PM2.5: Observation Uncertainty and Organic Tracer-Based Source Apportionment". In: *J. .Geophys. Res. Atmos.* 127 (5 2022), e2021JD035874. DOI: 10.1029/2021jd035874.

[7] J. Liu, M. Bergin, H. Guo, L. King, N. Kotra, E. Edgerton, and R. J. Weber. "Atmospheric Chemistry and Physics Size-resolved measurements of brown carbon in water and methanol extracts and estimates of their contribution to ambient fine-particle light absorption". In: *Atmos. Chem. Phys* 13 (2013), pp. 12389–12404. DOI: 10.5194/acp-13-12389-2013.

[8] R. Zhao, Q. Zhang, X. Xu, W. Wang, W. Zhao, W. Zhang, and Y. Zhang. "Light absorption properties and molecular compositions of water-soluble and methanol-soluble organic carbon emitted from wood pyrolysis and combustion". In: *Sci. Tot. Environ.* 809 (2022), p. 151136. DOI: 10.1016/j.scitotenv.2021.151136.

---

## Author Comment (AC2)

**Reviewer 2**

*Evans et al. investigated the effects of photochemical aging on biomass-burning organic aerosol (BBOA) under smouldering and flaming conditions. A new semi-quantitative UPLC-ESIMS workflow was used for the chemical characterisation of the fresh and aged samples. Overall, the study fits into the scope of Atmospheric Chemistry and Physics. However, some critical information is missing in the experimental designs and discussion, potentially lowering the scientific quality of the work. The manuscript requires major revision before it can be considered for publication.*

We thank the reviewer for reading our manuscript and providing some useful comments and suggestions. We have considered all the points made by the reviewer and we hope the following explanations are satisfactory in addressing the reviewer's comments.

**Major comments**

2.1 *Definition of atmospheric ageing: Please define "atmospheric ageing" in terms of oxidant exposure in the study. Please specify how OH radical was generated and how OH concentration was estimated in the chamber. What is the range of OH exposures or oxidant concentrations in the experiments?*

In our paper we define atmospheric aging as the combination of oxidation and dilution and evaporative processes as would occur in the real atmosphere to stove flue emissions. The authors will add further clarification of this into the manuscript in the experimental description. The chamber Xenon arc lamp and halogen lamp system in the Manchester Aerosol Chamber mimics atmospheric solar wavelengths enabling the photolysis of $NO_2$ to produce $O_3$. $O_3$ is then subsequently photolysed under these conditions to produce OH [1–3]. Heterogeneous wall chemistry will also produce HONO, which will photolyse to give OH and NO. The OH concentration inside the MAC has been reported previously as ca. $1\times10^6$ [1, 4] at similar $NO_x$ concentrations to these experiments.

"Each aging experiment, where aging is defined as being both photo-oxidative and dilution processes as would occur in the atmosphere upon emission from stoves, was repeated once."

2.2 *Line 139 to 141: How were the "flaming dominated" and "smouldering dominated" phases distinguished? Please specify what parameters you use. For example, modified combustion efficiency is a commonly used variable to differentiate flaming and smouldering. If possible, please include modified combustion efficiency in Table 1.*

During the experiments the burn phase was determined by controlling the oxygen content of the stove as described on line 146 to 148 and then using an expert visual assessment on the presence or lack of flame. However, MCE is a more robust metric and the MCE values now provided in Table 1 are in range for what is expected from smouldering and flaming phases.

Table 1: List of the OA samples used in this study and the initial conditions at the start of the aging period

| Experiment date | Conditions | Sample ID | Aging period / hrs | PM concentration / $\mu$g m$^{-3}$ | NO:NO$_2$ | OC:BC$^*$ | MCE |
|---|---|---|---|---|---|---|---|
| 21/04/2022 | Flaming light aged | FL_AGED_1 | 5:50 | 243.6 | 1.94 | 0.32 | 0.96 |
| 26/04/2022 | Smouldering light aged | SM_AGED | 6:05 | 213.6 | 1.81 | 406.3 | 0.78 |
| 28/04/2022 | Flaming light aged | FL_AGED_2 | 6:05 | 153.4 | 3.74 | 0.21 | 0.93 |
| 30/08/2022 | Flaming fresh flue | FL_FRESH | - | - | - | - | - |
| 31/08/2022 | Flaming fresh flue | SM_FRESH | - | - | - | - | - |

$^*$total organic content measured by AMS

2.3 *Lines 143 to 144: In the fresh experiments, the POA filter samples were directly taken from the wood burner flue without dilution. However, in the aged experiments, the POA sample was diluted after entering the chamber, potentially altering its chemical composition. Compounds with high volatilities will likely partition into the gas phase after dilution. Therefore, in terms of composition, the POA samples collected in fresh experiments were very likely to differ from those in aged experiments. In my opinion, a proper way to obtain a POA sample would be to sample the chamber before the lights are on. Also, the fresh (flue) and aged samples were obtained from experiments conducted over two different days. The manuscript does not provide evidence or data showing that the compositions of fresh emissions and/or combustion efficiencies were comparable over the two days (i.e., fresh vs. aged conditions). Therefore, it is unreasonable to compare the aging products observed by the ESI-Orbitrap-MS with the undiluted fresh emission samples. For example, in sections 3.2 and 3.3, the observed decreases in chemical compound concentrations in aged samples can be attributed to both the evaporation of initial POA materials after entering the chamber and the oxidation of POA after turning on the UV lights.*

The authors clarify that they observed good repeatability of the composition between experiments as shown in Figure 2 of our manuscript for the flaming experiment repeats. As stated in our comments above we define atmospheric aging as all oxidative, evaporative and dilution processes as would occur in the real atmosphere after a plume from the wood burning stove flue is emitted to the atmosphere. In our manuscript on line 253-255 we already mention that the aging processes in these experiments can also include evaporation from POA. Furthermore, given that the PM concentration in the chamber was 200 $\mu$g m$^{-3}$ or less at the start of the experiment, it was not possible to collect a filter with sufficient mass from the chamber before turning on the lights for the Orbitrap analysis.

**Minor comments**

2.4 *Line 35 to 37: Is there any more up-to-date information on the solid fuel percentage of POA in London? The information was more than ten years ago, and wood-burning activities in 2020 were compared with information from 2010.*

Unfortunately there are limited studies on the contribution of wood burning activities in the UK to POA. The only other similar study found by the authors was conducted in Dublin, Ireland during 2016 which observed solid fuel combustion activity contributed to 48-50% of OA [5]. We can however provide the reviewer with the most recent study of the contribution of wood burning to total PM rather than POA, which reports the contribution of wood burning in London during 2022 was 7-9% and 4-6% to PM$_{2.5}$ and PM$_{10}$ respectively [6]. In 2022 The Department for Food, Environment and Rural Affairs also reported domestic burning as the largest source (29%) of PM$_{2.5}$ in the UK [7]. This has been added to the text.

"In the UK, approximately 8 % of the population burn wood indoors (Department for Environment, Food and Rural Affairs, 2020 [8]) and in London solid fuel emissions comprised approximately 7-9% of PM$_{2.5}$ emissions during 2022 (Casey et al., 2023 [6]) and 26% of total primary OA (POA) during cold weather conditions consistent with domestic heating activity (Allan et al., 2010 [9])."

2.5 *Lines 105 – 107: The sentence "This semi-quantitative methodology had... in this study" should be moved into either the method or discussion part.*

We will move this into the experimental section as suggested.

2.6 *Table 1. What were the temperatures and relative humidities inside the chamber for each experiment? How did the author control the temperature and relative humidities and ensure they are similar among different experiments?*

The relative humidity was controlled between 50-60% and the temperature around 25 $^\circ$C. Temperature is controlled by an air conditioning system inside the chamber housing and relative humidity is controlled by an in-house built programmable logic controller (PLC)

board. The PLC board controls the valve system of the MAC for procedures such as humidification and fill/flush cycles etc. For further details on the Manchester Aerosol Chamber we recommend consulting the following papers [1, 2, 10]. We have added this information to the experiment section.

"The relative humidity throughout the experiment was controlled between 50-60 % and the temperature inside the chamber was kept around 25 ° C "

2.7 *Line 157 to 158: What is the pore size of the filter? Will a flow rate of 3 m3 min-1 cause filter breakthroughs or reduce the collection efficiency of the OA sample?*

The pore size is 2.2 $\mu$m. Many campaigns have been conducted at the Manchester Aerosol Chamber collecting SOA on Whatman 47mm Quartz QMA filters using the same collection method as this study with none of the above reported issues to our knowledge.

2.8 *Line 157 to 158: How long was the transportation process? How did the author ensure that filters were not contaminated during transportation? Please specify whether blanks were prepared to correct for the potential contamination of the samples.*

The filters were transported between the University of Manchester and the University of York which can be commuted between in a matter of hours. Filters were instantly stored in a foil pocket in a freezer at Manchester after collection, all handling of filters and pockets used gloves and clean utensils to reduce handling contamination. For transportation the filter pockets were placed within a ziplock plastic bag in a plastic container containing ice-packs to keep the filters at a cool temperature whilst out of the freezer. The container was then kept in an insulated bag or box whilst being transported. Blank filters of the chamber taken at the same time as the experiments and stored in the same conditions were subtracted from the biomass burning samples as explained on line 214-215.

2.9 *Line 165: How was the OC measured here? Does that refer to the total organic content measured by the AMS? If so, please correct the terminology (non-refractory organic) here and elsewhere to avoid misunderstanding.*

This is measured as the organic fraction from the AMS, the authors will correct the terminology in the text.

2.10 *Line 210: Please provide proper citations when using third-party software.*

Thank you for suggesting this we will add a reference [11, 12]

"In brief this method uses a non-target workflow developed in MZmine 2.53 and MZmine 3.9.0 software to detect features (Pluskal et al., 2010 [11]; Schmid et al., 2023 [12])"

2.11 *Line 214: Apart from elemental ratios, does the author have constraints on double bond equivalents? If so, please specify it here as well.*

Yes the default ring double bond equivalents in the MZmine formula prediction were applied which allows only positive and integer DBE values up to 40. However given our elemental criteria it is unlikely to reach such values. Furthermore, our analytical workflows are given in our methodology paper [13].

2.12 *Lines 244- 252: The effort of comparing findings between studies is appreciated. To strengthen the discussion, I recommend including literature data in Figure 1 for comparison.*

The authors will update Figure 1 to the following plot

[Figure]

2.13 *Lines 258-259: It is unclear why the conversion of NO to NO2 can indicate the oxidation of VOCs here. More background information needs to be provided for clarification.*

The authors do not believe further clarification is required in the manuscript but we have provided an explanation for the reviewer. As a VOC is oxidised by an OH radical to produce peroxy radicals ($RO_2$), NO is converted to $NO_2$ through the reactions of NO with $RO_2$. This then produces an alkoxy radical (RO) which can go on to form oxidation products, including $HO_2$, which also converts NO to $NO_2$ forming OH. This then propagates the $HO_x$, $RO_x$ radical oxidation cycle. As shown in Figure A2a of the manuscript NO is being converted to $NO_2$ via VOC oxidation through the above cycle. Eventually this reaches a steady state hence $NO:NO_2 = 1$ in Figure A2b from the manuscript as $NO_2$ is photolysed to produce NO and $O_3$.

2.14 *Line 266 to 267: What observations in Figure 1 showed that the OA contains POA, oPOA and SOA? How can the author differentiate the above three species using AMS data? Was positive matrix factorisation used for separation? Please clarify.*

We did not use a positive matrix factorisation for this separation. We observe the conversion of NO to $NO_2$ from VOC oxidation resulting in the formation of secondary gas and aerosol phase products. However, there will also be POA oxidation occurring at the same time as gas phase oxidation. Budisulistiorini et al., 2021 [14] show that the fractional amounts of $m/z$ 44 and $m/z$ 60 are indistinguishable between oPOA and SOA. Therefore, we assume oxidative photochemistry is occurring forming both oPOA and SOA, but we will add this reference to support this assumption to readers. It is also likely that there may be some unoxidised organic material left over after oxidation hence the inclusion of POA. From sections 3.2.1, in our analysis we see the retention of species observed in the POA in the aged samples (line 335-337) which supports our earlier assumption.

"Overall these results indicate the oxidation of POA from fresh domestic BB emissions to form oxidised POA (oPOA) and SOA which are indistinguishable with respect to $f_{44}$ and $f_{60}$ (Budisulistiorini et al., 2021 [14])."

2.15 *Lines 283 to 284: Eq. 2 and 3 differ from the equations stated in the cited reference and its erratum (Koch and Dittmar, 2006). Please check whether the current study's calculations were based on the correct equations. Please specify if these equations were modified based on the original ones.*

The equations refer to those in the corrected article published in 2016 [15], we will update the referencing in the text on Line 294.

2.16 *Figures 2 and 3: The colour code for different DBEs is confusing and reduces the readability of the figure. Please consider grouping the DBEs into several categories, as the significance of reporting DBEs individually is unclear.*

The authors believe the colour scale in the figure is readable and sufficiently different to clearly show the regions of lower DBE (lower C numbers) and higher DBE (high C numbers). However, if the editor feels it is also unclear then we can change the colour scheme. We report individual DBEs as it allows us to see how many unsaturated functional groups may be represent on an aromatic ring (ie. 4 for the ring + n for n substituent) and the presence of dimers. For example in Figure 2 showing fresh emissions at $C_8$ there is a large contribution of DBE 5 and at $C_{16}$ there is a contribution of DBE 10 species which could be potential dimers of the $C_8$ compound. Consulting the same figure we can see from fresh emission $C_8$ is mostly DBE 5 but upon aging this increases to DBE 6 indicating the addition of an oxidised functional group. By grouping the DBE we would not be able to see such compositional differences and details at this granular level.

2.17 *Line 338 to 339: Please confirm whether these two equations are correct. If they differ from Eq. 2 and 3, please use other terms to describe DBEAI and CAI expressed in Eq. 5 and 6.*

These equations are different, please see our response to comment 1.4. We will however change the labelling in the text to reduce any confusion.

"

$$DBE_{\text{CHON}} = 1 + C - \frac{O-2}{2} - S - \frac{H}{2} - \frac{N-1}{2} \tag{1}$$

$$C_{\text{CHON}} = C - \frac{O-2}{2} - S - (N-1) \tag{2}$$

$$AI_{\text{CHON}} = \frac{DBE_{\text{CHON}}}{C_{\text{CHON}}} \tag{3}$$

"

2.18 *Most of the results in Sections 3.2 and 3.3 agree with the literature. What are the novelties of the current study? If the results are similar, what is the advantage of using the semi-quantitative workflow suggested by the current research to investigate photooxidation reactions?*

This study is the first reported quantitative non-target analysis applied to biomass burning aerosol and the first semi-quantification to incorporate retention window scaling with over 100 surrogate standards. Thereby this non-target analysis is the first to account for ionisation efficiency compared to previous studies on biomass burning aerosol (eg. [16, 17]) enabling more robust estimates of species concentration and therefore the elemental composition. As mentioned in the manuscript (see line 67-71) there are numerous advantages of using non-target analysis over targeted analysis for the exploration of chemical composition given the sheer complexity of the samples [13]. For example, we detected up to 2357 features in a single sample which is simply impossible to infer the composition from a targeted approach with a limited number of standards ($\approx 100$). A significant novel finding is the dominance of CHO species whereas previous studies observe high contributions of CHON species, predominantly due to the use of peak area as a metric of abundance and these compounds typically possess high ionisation efficiencies. We have compared our results to literature for the discussion of potential identities to the compounds detected by our non-target workflow. Overall, we believe this work shows the wider picture of the OA composition and the crucial role of PAHs during aging in the determination of toxicity, neither of which have been observed in an Orbitrap study of wood burning aerosol before.

2.19 *Since AMS data is available, I would appreciate it if an analysis of organic nitrates could be carried out. This information will potentially help explain the changes in CHON compounds during aging.*

The authors agree with the reviewer that this analysis will be of interest however the suggested work is currently being done by one of our co-authors and will be published in a subsequent companion paper. Furthermore, our technique is highly selective towards nitrophenols whereas an AMS is for organic nitrates. In this work we found a relatively minor contribution of CHON species ($\leq 10\%$), likely to be nitrophenols, therefore the majority of the analysis is centered towards CHO species. We believe the figures provided detail the

aging of the CHON species detected by the Orbitrap analysis with Figure 4 and 5 showing that in the flaming phase there is production of CHON species whereas for smouldering there is an overall loss which is most evident for polyaromatic CHON species.

**Technical Comments**

The following technical comments have been addressed in the main text. Where further clarification was required these are shown in blue.

1. *Legends and texts in figures in the main text and appendix are currently too small to read. Please increase them for readability*

2. *Label the subfigure with letters "a), b), c), etc." so that it is clear which subfigure is referred to in the main text.*

3. *Line 128: It should be "6 kW".*

4. *Line 133: Harsher to what? What was compared with?* Harsher to the overnight clean described in the previous sentence.

5. *Line 151: Please specify what had been injected in the sentence "Following injection...".*

6. *Lines 153 – 155: The sentence "The smoke was aged for approximately 6 hours... for offline chemical composition analysis" is difficult to follow. Please rewrite it.*

7. *Line 167 to 168: The experiment dates shown in Table 1 are April and August. Please check if the "September campaign" was a typo or if it refers to something else.*

8. *Line 174: It is supposed to be "$SO_4^{2-}$".*

9. *Line 183: Higher aerosol mass loading to what? What was compared with?* This refers to the higher mass loading of the fresh flue samples compared to those from the chamber which uses the entire filter in the extraction.

10. *Lines 223 – 225: Please use a table or figure to summarise the scaling factor for each window.* The scaling factors used here are instrument specific and we would recommend future users of the method to perform their own necessary calibrations. We can however refer readers to our methodology paper where this information can be found.

11. *Figure A1 a: NO3-*

12. *Figure A1 b: Please describe what the inset plot refers to in the caption.* It shows the zoomed in BC timeseries for smouldering as indicated by the colour and marker shape

13. *Figure 1: Please provide legends of the figure. For example, it's unclear what the difference is between the circle and square points. Please use different marker styles or separate them into three sub-figures. It's hard to distinguish the three marker types in Figure 1.*

**Bibliography**

[1] A. Voliotis, Y. Wang, Y. Shao, M. Du, T. J. Bannan, C. J. Percival, S. N. Pandis, M. R. Alfarra, and G. Mcfiggans. "Exploring the composition and volatility of secondary organic aerosols in mixed anthropogenic and biogenic precursor systems". In: *Atmos. Chem. Phys* 21 (2021), pp. 14251–14273. DOI: 10.5194/acp-21-14251-2021.

[2] A. Voliotis, M. Du, Y. Wang, Y. Shao, M. R. Alfarra, T. J. Bannan, D. Hu, K. L. Pereira, J. F. Hamilton, M. Hallquist, T. F. Mentel, and G. Mcfiggans. "Chamber investigation of the formation and transformation of secondary organic aerosol in mixtures of biogenic and anthropogenic volatile organic compounds". In: *Atmos. Chem. Phys* 22 (2022), pp. 14147–14175. DOI: 10.5194/acp-22-14147-2022.

[3] Y. Shao, A. Voliotis, M. Du, Y. Wang, K. Pereira, J. Hamilton, M. R. Alfarra, and G. Mcfiggans. "Chemical composition of secondary organic aerosol particles formed from mixtures of anthropogenic and biogenic precursors". In: *Atmos. Chem. Phys* 22 (2022), pp. 9799–9826. DOI: 10.5194/acp-22-9799-2022.

[4] A. Voliotis, M. Du, Y. Wang, Y. Shao, T. J. Bannan, M. Flynn, S. N. Pandis, C. J. Percival, M. R. Alfarra, and G. McFiggans. "The influence of the addition of isoprene on the volatility of particles formed from the photo-oxidation of anthropogenic–biogenic mixtures". In: *Atmos. Chem. Phys.* 22 (20 2022), pp. 13677–13693. DOI: 10.5194/acp-22-13677-2022.

[5] C. Lin, D. Ceburnis, R.-J. Huang, W. Xu, T. Spohn, D. Martin, P. Buckley, J. Wenger, S. Hellebust, M. Rinaldi, M. C. Facchini, C. O'Dowd, and J. Ovadnevaite. "Wintertime aerosol dominated by solid-fuel-burning emissions across Ireland: insight into the spatial and chemical variation in submicron aerosol". In: *Atmos. Chem. Phys.* 19 (22 2019), pp. 14091–14106. DOI: 10.5194/acp-19-14091-2019.

[6] J. Casey, L. Mittal, L. Buchanan, G. Fuller, and I. Mead. *London wood burning project: air quality data collection.* Tech. rep. Environmental Research Group, Imperial College London, 2023.

[7] DEFRA. *Emissions of air pollutants in the UK – Particulate matter (PM10 and PM2.5) - GOV.UK.* Tech. rep. Department for Environment, Food and Rural Affairs, 2024.

[8] Department for Environment Food & Rural Affairs (DEFRA). *Burning in UK Homes and Gardens Research Report.* Tech. rep. DEFRA, 2020.

[9] J. D. Allan, P. I. Williams, W. T. Morgan, C. L. Martin, M. J. Flynn, J. Lee, E. Nemitz, G. J. Phillips, M. W. Gallagher, and H. Coe. "Contributions from transport, solid fuel burning and cooking to primary organic aerosols in two UK cities". In: *Atmos. Chem. Phys* 10 (2010), pp. 647–668. DOI: 10.5194/acp-10-647-2010.

[10] Y. Shao, Y. Wang, M. Du, A. Voliotis, M. R. Alfarra, S. P. O'Meara, S. F. Turner, and G. McFiggans. "Characterisation of the Manchester Aerosol Chamber facility". In: *Atmos. Meas. Tech.* 15 (2 2022), pp. 539–559. DOI: 10.5194/amt-15-539-2022.

[11] T. Pluskal, S. Castillo, A. Villar-Briones, and M. Orešič. "MZmine 2: Modular framework for processing, visualizing, and analyzing mass spectrometry-based molecular profile data". In: *BMC Bioinformatics* 11 (1 2010), pp. 1–11. DOI: 10.1186/1471-2105-11-395.

[12] R. Schmid, S. Heuckeroth, A. Korf, A. Smirnov, O. Myers, T. S. Dyrlund, R. Bushuiev, K. J. Murray, N. Hoffmann, M. Lu, A. Sarvepalli, Z. Zhang, M. Fleischauer, K. Dührkop, M. Wesner, S. J. Hoogstra, E. Rudt, O. Mokshyna, C. Brungs, K. Ponomarov, L. Mutabdžija, T. Damiani, C. J. Pudney, M. Earll, P. O. Helmer, T. R. Fallon, T. Schulze, A. Rivas-Ubach, A. Bilbao, H. Richter, L. F. Nothias, M. Wang, M. Orešič, J. K. Weng, S. Böcker, A. Jeibmann, H. Hayen, U. Karst, P. C. Dorrestein, D. Petras, X. Du, and T. Pluskal. "Integrative analysis of multimodal mass spectrometry data in MZmine 3". In: *Nat. Biotechnol.* 41 (4 2023), pp. 447–449. DOI: 10.1038/s41587-023-01690-2.

[13] R. Evans, D. Bryant, A. Voliotis, D. Hu, H. Wu, S. Syafiraa, O. Oghama, G. McFiggans, J. Hamilton, and A. Rickard. "A Semi-Quantitative Approach to Nontarget Compositional Analysis of Complex Samples". In: *Anal. Chem.* 96 (2024), pp. 18349–18358. DOI: 10.1021/acs.analchem.4c00819.

[14] S. H. Budisulistiorini, J. Chen, M. Itoh, and M. Kuwata. "Can Online Aerosol Mass Spectrometry Analysis Classify Secondary Organic Aerosol (SOA) and Oxidized Primary Organic Aerosol (OPOA)? A Case Study of Laboratory and Field Studies of Indonesian Biomass Burning". In: *ACS Earth Space Chem.* 5 (12 2021), pp. 3511–3522. DOI: 10.1021/acsearthspacechem.1c00319.

[15] B. P. Koch and T. Dittmar. "From mass to structure: an aromaticity index for high-resolution mass data of natural organic matter". In: *Rapid Commun. Mass Spectrom.* 30.1 (2016), pp. 250–250. DOI: 10.1002/rcm.7433.

[16] K. Dzepina, C. Mazzoleni, P. Fialho, S. China, B. Zhang, R. C. Owen, D. Helmig, J. Hueber, S. Kumar, J. A. Perlinger, L. J. Kramer, M. P. Dziobak, M. T. Ampadu, S. Olsen, D. J. Wuebbles, and L. R. Mazzoleni. "Molecular characterization of free tropospheric aerosol collected at the Pico Mountain Observatory: A case study with a long-range transported biomass burning plume". In: *Atmos. Chem. Phys.* 15 (9 2015), pp. 5047–5068. DOI: 10.5194/acp-15-5047-2015.

[17] M. Brege, M. Paglione, S. Gilardoni, S. Decesari, M. C. Facchini, and L. R. Mazzoleni. "Molecular insights on aging and aqueous-phase processing from ambient biomass burning emissions-influenced Po Valley fog and aerosol". In: *Atmos. Chem. Phys.* 18 (17 2018), pp. 13197–13214. DOI: 10.5194/acp-18-13197-2018.

---

## Author Response (AR2)

**Reviewer 2**

We thank the reviewer for reading our revised manuscript and providing additional insightful suggestions. We hope that the following responses and additions to the manuscript address these comments.

2.1 Response to comment 2.3: *Figure 2 only shows the composition of the repeated aging experiment. Please clarify whether the compositions of the fresh emission for the flaming experiments (FL_AGED_1, FL_AGED_2 and FL_FRESH) are comparable or not and how the similarity was examined. This is because when the authors were comparing fresh and aged samples, these samples were obtained from two different burning experiments (e.g. FL_AGED_1 on 21/04/2022 vs. FL_FRESH on 30/08/2022). Therefore, it is vital to ensure that the chemical composition of the fresh emission in FL_AGED_1 is similar to FL_FRESH. I believe that without this crucial information, the data quality would not fit into the very high standards of the Journal of Atmospheric Chemistry and Physics. Moreover, the observed good repeatability in Figure 2, claimed by the author, is somewhat subjective. I found observable differences between aged samples from the two flaming experiments. For example, the second most dominating species are C11 compounds with DBE = 0 in FL_AGED_1, while the second most dominating species in FL_AGED_2 are C10 compounds. Please clarify how the similarity between repeated aging experiments was examined in the main text.In addition, the difficulty of collecting the diluted fresh samples from the chamber, which was mentioned in the response, must be acknowledged in the main text.*

The authors previously mentioned in the original manuscript on line 167 that the aerosol was exclusively sampled from the chamber in the first campaign, therefore, there are no direct comparisons with the flue. The emissions from the FL_FRESH sample on 30/08/2022 were aged in the chamber, however, the sampling of the fresh emissions occurred separately to the filling of the chamber and proceeded after the emissions were injected. Furthermore, this aging experiment had a lack of several key gas phase measurements which meant we chose not to include it in the paper. Despite this, we do see the same overarching trends as in Figure 2 with a peak at $C_8$, fragmentation and loss of aromaticity including significant reductions of the polyaromatic CHO content. The authors do acknowledge that in burning experiments it can be difficult to achieve the exact chemical composition each time due to the complexity of lignin breakdown in a fire and the naturally existing small differences between different pieces of wood even from the same source. However, we disagree with the repeatability of our results as we show that the overall trends are replicated in the repeats. In Figure 2, we show that the general overarching trend in fragmentation to $C_7$-$C_{10}$ compounds with a peak at $C_8$ and loss of aromaticity upon aging is consistent across the flaming aged repeats despite small differences such as those mentioned by the reviewer. We also refer the reviewer to Figure 1 showing the $f_{44}$ vs $f_{60}$ space measured by AMS, where it is clear the flaming burns are similar in their initial values and temporal evolution upon aging as these points occupy the same area of space. Furthermore in Figures 4 and 5 the flaming aged burns have very similar O:C probability density functions and occupy the same H:C and O:C space in the Van Krevelen diagram. Therefore, we believe our experiments to be repeatable across the entire bulk chemical composition despite some small differences in individual compounds and therefore the comparisons made between the fresh and aged emissions to be acceptable. We will include that we could not sample from the chamber directly due to mass constraints as we mention in our previous responses. We have also added a sentence to the conclusions to indicate that the experiment methodology could be improved by sampling the fresh emissions at the start of the experiment from the chamber but that this remains challenging due to the low mass present and the need to sample large volumes for offline approaches (See comment 2.4).

"Page 6 line 150: Due to the relatively low particle concentration of 200 $\mu g$ m$^{-3}$ inside the chamber the fresh aerosol was sampled directly from the flue of the wood burner to yield sufficient mass for offline chemical compositional analysis rather than from the chamber itself. POA was sampled from the flue at 2 L min$^{-1}$ for 5 minutes."

2.2 Generation of OH: *The author gave explicit details about how OH was produced in the chamber in response to comment 2.1. The same levels of detail need to be included in the main text to clarify OH production and the estimated OH concentrations.*

The authors will add this explanation to the existing description of the chamber lighting for the aging experiments on page 5 line 126.

"The chamber was illuminated using two 6 kw xenon arc lamps with quartz fibre glass filters and 4 rows of halogen lamps (64 bulbs) to simulate atmospheric solar wavelengths, which enables the photolysis of $NO_2$ to produce $O_3$. $O_3$ is then subsequently photolysed in the presence of water molecules to produce OH [1–3]. Heterogeneous wall chemistry will also produce HONO, which is photolysed to yield OH and NO. The OH concentration inside the MAC has been previously calculated as ca. $1\times10^6$ molecules $cm^{-3}$ [1, 4] at similar $NO_x$ concentrations to these experiments."

2.3 Lines 265 – 266: *Thanks for the revision. Given that oxidised POA (oPOA) and SOA are indistinguishable using f44 and f60, we cannot rule out the possibility that the oxidation of POA from fresh domestic BB emissions only forms oPOA. If so, please revise the sentence.*

The authors disagree with this statement, it is highly unlikely that SOA is not formed during the oxidation and it is well known that SOA is formed upon OH exposure of biomass burning emissions in chamber studies. We refer the reviewer to a previous study of SOA formation from the oxidation of biomass burning emissions where an increase in SOA mass measured by AMS was observed in all 20 aging experiments during the FIREX chamber study [5].

2.4 Response to Comment 2.18: *Thanks for your explanation. I recommend that the authors include similar levels of detail in the abstract and/or conclusion to highlight the importance and novelty of the current study*

We have adapted some of our statements in the abstract and conclusion as shown below.

"**Abstract line 19:** This study presents the first reported quantitative non-target compositional analysis of domestic BBOA using retention window scaling and demonstrates compositional changes between burn phase and after aging may have important consequences for exposure to such emissions in residential settings. **Conclusion line 503:** The chemical composition of domestic biomass burning organic aerosol (BBOA), from a series of controlled burn chamber experiments, was investigated using a newly developed semi-quantitative non-target analysis (NTA) methodology and is the first study to account for ionisation effects in an NTA of BBOA using retention window scaling [6]. The NTA approach enabled the detection of up to 2357 features in a single sample [6] which is simply impossible from a targeted approach with a limited number of standards. Significant compositional differences between the organic aerosol (OA) derived from emission under different burn phases (i.e. flaming dominated and smouldering dominated phases) and after aging were observed. However, the experimental methodology used here could be improved by sampling directly from the chamber at the start of the experiment but this remains challenging for offline approaches due to the higher sample volume required and the low particle mass used in this study. Overall, the composition of domestic BBOA was dominated by oxygenated compounds (CHO). On average CHO compounds constituted 90% of the total detected mass with a smaller contribution ( $\leq 10\%$) of organonitrogen species (CHON), which suggests the wide usage of nitroaromatic compounds as tracers of biomass burning may not be ideal as these compounds typically have high ionisation efficiencies and therefore appear more abundant when using peak area metrics."

2.5 Response to Comment 2.19: *The additional analysis of organic nitrates (ON) from AMS is straightforward. Comparing ON between AMS and Orbitrap measurements is potentially interesting to the community. If the editor feels this will be too much work, the author can leave this one out.*

As stated in our previous response this work is being carried out by one of our co-authors and will appear in a separate companion publication. The main findings of this work is the non-target analysis of the high resolution Orbitrap mass spectra and we believe it is beyond the scope of the current manuscript to include the additional AMS analysis.

**Bibliography**

[1]  A. Voliotis, Y. Wang, Y. Shao, M. Du, T. J. Bannan, C. J. Percival, S. N. Pandis, M. R. Alfarra, and G. Mcfiggans. "Exploring the composition and volatility of secondary organic aerosols in mixed anthropogenic and biogenic precursor systems". In: *Atmos. Chem. Phys* 21 (2021), pp. 14251–14273. DOI: 10.5194/acp-21-14251-2021.

[2]  A. Voliotis, M. Du, Y. Wang, Y. Shao, M. R. Alfarra, T. J. Bannan, D. Hu, K. L. Pereira, J. F. Hamilton, M. Hallquist, T. F. Mentel, and G. Mcfiggans. "Chamber investigation of the formation and transformation of secondary organic aerosol in mixtures of biogenic and anthropogenic volatile organic compounds". In: *Atmos. Chem. Phys* 22 (2022), pp. 14147–14175. DOI: 10.5194/acp-22-14147-2022.

[3]  Y. Shao, A. Voliotis, M. Du, Y. Wang, K. Pereira, J. Hamilton, M. R. Alfarra, and G. Mcfiggans. "Chemical composition of secondary organic aerosol particles formed from mixtures of anthropogenic and biogenic precursors". In: *Atmos. Chem. Phys* 22 (2022), pp. 9799–9826. DOI: 10.5194/acp-22-9799-2022.

[4]  A. Voliotis, M. Du, Y. Wang, Y. Shao, T. J. Bannan, M. Flynn, S. N. Pandis, C. J. Percival, M. R. Alfarra, and G. McFiggans. "The influence of the addition of isoprene on the volatility of particles formed from the photo-oxidation of anthropogenic–biogenic mixtures". In: *Atmos.Chem. Phys.* 22 (20 2022), pp. 13677–13693. DOI: 10.5194/acp-22-13677-2022.

[5]  C. Y. Lim, D. H. Hagan, M. M. Coggon, A. R. Koss, K. Sekimoto, J. de Gouw, C. Warneke, C. D. Cappa, and J. H. Kroll. "Secondary organic aerosol formation from the laboratory oxidation of biomass burning emissions". In: *Atmospheric Chemistry and Physics* 19 (19 2019), pp. 12797–12809. DOI: 10.5194/acp-19-12797-2019.

[6]  R. Evans, D. Bryant, A. Voliotis, D. Hu, H. Wu, S. Syafira, O. Oghama, G. McFiggans, J. Hamilton, and A. Rickard. "A Semi-Quantitative Approach to Nontarget Compositional Analysis of Complex Samples". In: *Anal. Chem.* 96 (2024), pp. 18349–18358. DOI: 10.1021/acs.analchem.4c00819.